# Sparsh: Self-supervised touch representations for vision-based tactile sensing

Carolina Higuera[1,2*], Akash Sharma[1,3*], Chaithanya Krishna Bodduluri[1],
Taosha Fan[1], Patrick Lancaster[1], Mrinal Kalakrishnan[1], Michael Kaess[3], Byron Boots[2],
Mike Lambeta[1], Tingfan Wu[1], Mustafa Mukadam[1]

[1]FAIR at Meta, [2]University of Washington, [3]Carnegie Mellon University
*Equal contribution

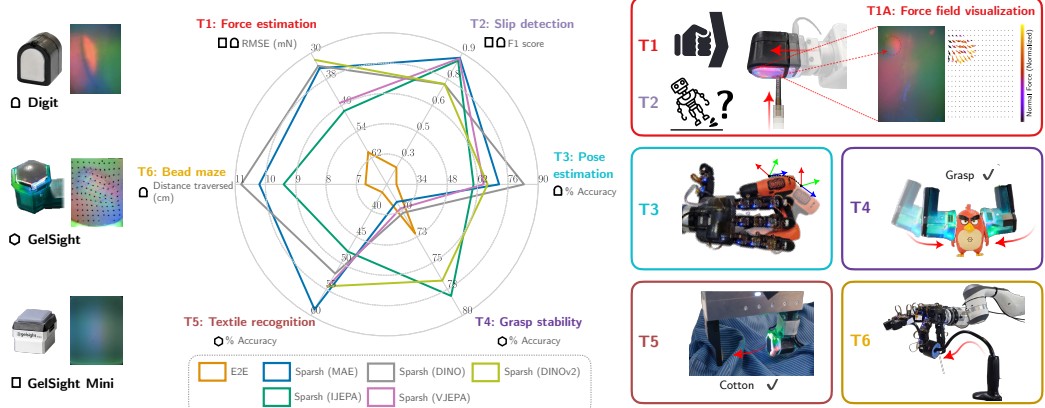

**Figure 1:** We present Sparsh, a family of general touch representations, and TacBench, a standardized benchmark of six touch-centric tasks ([T1]-[T6]) covering prominent problems in vision-based tactile sensing. We find Sparsh pre-trained with self-supervision on a dataset of 460k+ tactile images can generalize across many tasks (right) and sensors (left) outperforming task and sensor specific models (E2E). Performance in the plot (middle) is with task decoders using 33% labeled data (except [T6] that uses 50%).

**Abstract:** In this work, we introduce general purpose touch representations for the increasingly accessible class of vision-based tactile sensors. Such sensors have led to many recent advances in robot manipulation as they markedly complement vision, yet solutions today often rely on task and sensor specific handcrafted perception models. Collecting real data at scale with task centric ground truth labels, like contact forces and slip, is a challenge further compounded by sensors of various form factor differing in aspects like lighting and gel markings. To tackle this we turn to self-supervised learning (SSL) that has demonstrated remarkable performance in computer vision. We present Sparsh, a family of SSL models that can support various vision-based tactile sensors, alleviating the need for custom labels through pre-training on 460k+ tactile images with masking and self-distillation in pixel and latent spaces. We also build TacBench, to facilitate standardized benchmarking across sensors and models, comprising of six tasks ranging from comprehending tactile properties to enabling physical perception and manipulation planning. In evaluations, we find that SSL pre-training for touch representation outperforms task and sensor-specific end-to-end training by 95.1% on average over TacBench, and Sparsh (DINO) and Sparsh (IJEPA) are the most competitive, indicating the merits of learning in latent space for tactile images. Project page: https://sparsh-ssl.github.io/

**Keywords:** Tactile sensing, Pre-trained representations, Self-supervised learning

## 1 Introduction

Touch comes before sight, before speech. In today's AI landscape, this Margaret Atwood quote is playing out in reverse despite touch being a crucial modality for humans to physically interact

8th Conference on Robot Learning (CoRL 2024), Munich, Germany.

with the world. Touch provides a direct window into information like forces and contact during hand-object interactions, enabling dexterity. Vision-based tactile sensors [1, 2, 3, 4] have emerged as the leading form factor capable of capturing images of physical interactions at the sensor-object-environment interface, often inaccessible through vision. These images contain properties such as contact geometry, texture, and forces and have been leveraged across tasks like insertion [5, 6], pushing [7], grasping [8], localization [9], and pose and shape estimation [10, 11].

The prevailing approach to incorporating vision-based tactile sensors in robot tasks is to train custom models using labeled data [6, 12, 13, 14] to estimate useful states. However, this can be inefficient and results in repeated effort across different type of sensors like GelSight 2017 [1] (with markers) and DIGIT [3] (without markers) or different variety of tasks. For example, feature extractors trained on GelSight with markers may not transfer to other sensors, and encoders optimized for texture recognition [15] may not be suitable for tasks that require reasoning about forces or slip [16]. Supervision for building large general models is prohibitive as collecting large scale real world data with ground truth labels is challenging. For instance, properties like forces [17] and slip [18] require careful and expensive instrumentation in lab settings, while other properties like tracking deformations [19] or extrinsic contact [6] can be infeasible. To address this fragmentation in the literature across custom solutions, there is a need for touch representations that are broadly applicable to many tasks and many sensors, along with a benchmark of standardized tasks useful in measuring progress. Taking inspiration from self-supervised learning (SSL) methods in computer vision, we extend these approaches to the tactile sensing domain and build a benchmark for evaluation (Figure 1).

In this work, we introduce a family of touch representations for vision-based tactile sensors trained with SSL. Specifically, we provide a recipe to adapt masking-based objectives from computer vision to the tactile domain, and train general-purpose touch encoders by curating a new Touch-Slide dataset and existing datasets of tactile images (Figure 2), namely YCB-Slide [9], Touch-and-Go [20], and ObjectFolder [21]. Pulling together additional unlabeled data points from the existing datasets we train our models on a total of 460k+ tactile images. Finally, we construct TacBench, a benchmark consisting of six touch-centric tasks that cover the space of relevant problems on tactile properties such as force estimation and slip detection, on perception such as pose estimation and grasp stability, and on robot manipulation such as policies for solving a bead maze. Our contributions are as follows:

1. General touch representations, Sparsh pre-trained with SSL on 460k+ tactile images,
2. TacBench a benchmark of standardized tasks to evaluate touch representations and models, and
3. Curation of new & existing datasets, unlabeled for SSL and labeled for benchmarking.

In evaluations on TacBench, we find that Sparsh with SSL pre-training yield on average 95.1% improvement over task and sensor specific end-to-end models under limited labeled data budget (33%-50% of the collected amount) for any task. Additionally, we find Sparsh (DINO) and Sparsh (IJEPA) to be the most competitive outperforming Sparsh (MAE), indicating the merits of learning in latent space over pixel space for tactile images.

## 2   Related work

Self-Supervised Learning with its success in natural language processing and computer vision, has become the new learning paradigm. In the last three years, a variety of general-purpose frameworks [22, 23, 24, 25] have been proposed for learning representations. We refer to [26, 27] for a comprehensive survey on SSL frameworks and their categorization based on pretext tasks and learning algorithms. In Appendix B, we expand on Masked Image Modeling (MIM), self-distillation, and Joint-Embedding Predictive Architecture (JEPA), as we explore them in this study.

Traditionally, tactile sensing has relied on preprocessing tools like marker tracking and finite element method models to extract contact properties, such as shear forces [16, 28], dense normal estimation [29, 28], and contact area prediction [30]. From a learning perspective, a trend is to use custom encoder architectures tailored for specific tasks and sensors, which are either pre-trained or trained end-to-end [31, 12, 32, 13, 14, 6, 9]. Nevertheless, there is an increasing interest in representation learning for vision-based tactile sensors. For instance, MAE has shown effectiveness at material classification and texture recognition [33]. Fine-tuning convolutional encoders for BioTac, RoboSkin, and GelSight performs well on fabric decomposition tasks [15]. Even nearest-neighbor retrieval over

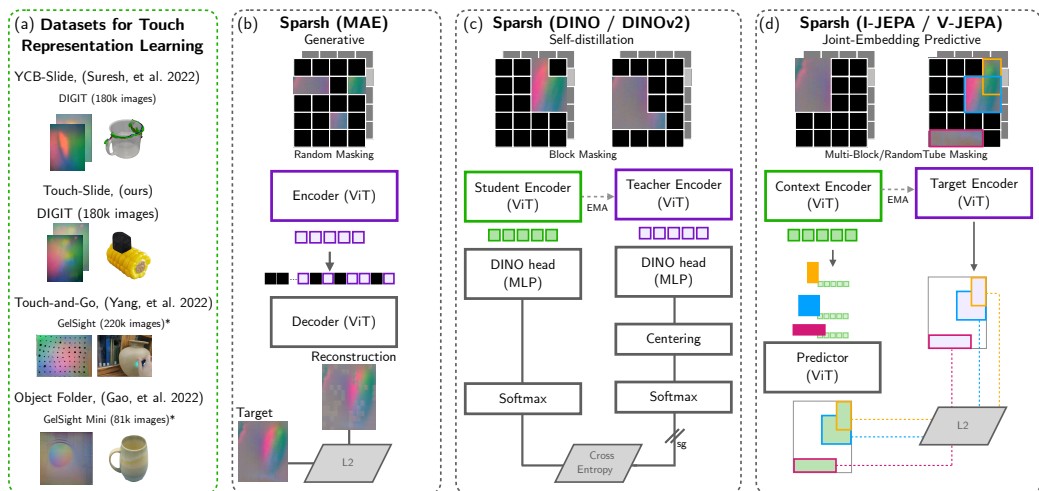

**Figure 2:** (a) We curate new and existing datasets of vision-based tactile sensors to train touch representations by adapting state-of-the-art SSL vision methods to the tactile domain, namely (b) Masked Autoencoder (MAE) [23], (c) DINO/DINOv2 [48, 49], and (d) Image/Video Joint-Embedding Predictive Architecture (JEPA) [50, 51]. *Without need for labels we can sample more images than reported in Touch-and-Go [20] and Object Folder [21].

pretrained representations, for the XELA [34] uskin sensor, can enable some success in dexterous manipulation [35]. Crucially, the current state of standardization in learning touch representations and the wide variety of tactile sensors available, has made it challenging to develop and share pre-trained models across the research community in this domain.

Another direction is exploring the alignment of visual and tactile modalities in latent space using multimodal datasets [20, 36, 37, 38, 8, 39] and techniques like contrastive coding and cross-sensory retrieval [40, 41, 42], yielding promising results in tasks like material classification, grasp stability, and tactile-driven image stylization. However, current approaches [40, 43, 44, 42] primarily focus on texture and visual properties and overlook physical contact properties, such as forces, slippage, and poses, which are essential for dexterous manipulation.

The works closest and concurrent to ours are T3 [45] and UniT [46]. T3 trains sensor-specific encoders to capture shared latent information through a shared trunk, using both the MAE objective and labeled task-specific data as supervision. UniT is a VQGAN [47] model with a patch-based discriminator for representation learning only for GelSight Mini (markers). On the other hand, we introduce a family of models trained with the latest SSL algorithms for the three most commonly used families of tactile sensors: DIGIT, GelSight 2017, and GelSight Mini. Similar to T3 and UniT, we evaluate touch representations for policy learning, however we also introduce a standardized benchmark to comprehensively evaluate representations and their ability to solve several relevant touch-centric tasks along tactile properties, physical perception, and manipulation planning.

## 3   Touch representations via self-supervised learning

Current approaches incorporating vision-based tactile sensors in robotic tasks lean on custom task and sensor specific solutions. As highlighted in the introduction, this can be inefficient, and there is a growing need for general-purpose touch representations that can be more broadly useful. We envision the following guiding principles for such general touch representations: (i) provide performance benefits across many tasks including real-time robot manipulation, (ii) generalize across multiple types of sensors built on a similar operating principle, like vision-based tactile sensors, and (iii) improve performance by leveraging computation and diverse data at scale without the need for manual labels. Self-supervised learning (SSL) is promising in this regard, as it offers data-agnostic objectives based on wide-reaching concepts such as analysis-by-synthesis to train generalist models. This motivates the question of whether vision techniques such as masked image modeling (MIM) [23, 50] and self-distillation [48, 49] can be extended to the domain of vision-based tactile sensors.

To this end, inspired from advances in self-supervised learning (SSL) in computer vision, we introduce Sparsh, a family of touch representation trained with SSL across multiple sensors such as DIGIT [3],

GelSight 2017 (with markers [1]), and GelSight Mini (without markers). The tactile domain, however, imposes several challenges that impede a straightforward application of SSL approaches from vision towards touch representations.

Tactile sensors inherently provide local information; thus, images can be ambiguous when observed independently and can vary across grasp forces, materials, and shapes. Therefore, we investigate the optimal space for training SSL encoders. Specifically, we are interested in the efficiency of pixel reconstruction, latent reconstruction, or clustering approaches to learning representations in the presence of aforementioned ambiguities. We hypothesize that latent reconstruction and clustering could be more efficient in learning representations, as they focus model capacity away from fine reconstruction details [52]. Tactile images contain distractors, such as markers and light placement variations, which can significantly vary due to manufacturing discrepancies. To increase robustness to distractors, we perform background subtraction for both DIGIT and GelSight Mini (markerless). This process provides the model with a reference to no-contact, which conveys static shear information when a perpendicular force is applied to the elastomer. We find empirically that background subtraction helps models generalize across the same type of sensor.

To address the scarcity of labeled and even unlabeled data in the tactile domain that limits the training of large encoder models, we curate together new and existing vision-based tactile sensor datasets [9, 20, 21], totaling $\sim$661k images as illustrated in Figure 2. 70% or 462.7k images are used for SSL pre-training which is an order of magnitude larger than any prior work on touch representations [33, 40, 43] (the rest are held out for monitoring training using online probes).

Tokenizing tactile images appropriately for SSL is important because many tasks such as slip detection and relative pose estimation require temporal reasoning. For SSL methods that operate on images, we concatenate two tactile images with a temporal stride of 5 samples across the channel dimension, $I_t \oplus I_{t-5} \to x \in \mathbb{R}^{h \times w \times 6}$. For a sensor operating at 60FPS, this corresponds to an inference window of approximately $80\ ms$, the reaction time that humans need to adjust the grip force when detecting partial slip [53]. For SSL methods that operate on video (e.g. V-JEPA), we generate clips with 4 frames at $[t, t-2, t-4, t-6] \in \mathbb{R}^{4 \times h \times w \times 3}$ corresponding to an inference window of $\sim 100\ ms$. Currently Sparsh is limited by data streaming rates, and not by inference time, as the models support inference rates of upto 112FPS (measured on an Nvidia RTX3080). See Appendix C for additional details on model architectures and training.

## 4   TacBench: Tactile sensing benchmark

We introduce TacBench, a collection of touch-centric tasks, and labeled datasets for standardized evaluation for vision-based tactile sensing. We compile data for all tasks from various sensors to evaluate the generalization of representations. These tasks are categorized under three main questions.

**1. Do the representations comprehend tactile properties?** Tactile sensing informs finger-object contact interaction properties like forces and slip that are crucial for robot manipulation. In Section 5, we evaluate learned representation on estimating instantaneous normal and shear forces [T1] and visualizing force fields [T1A] [16, 28, 29, 17], and detecting slip [T2] [16, 54, 55, 18].
**2. Do the representations enable perception?** Tracking and accumulating slip states is essential for tasks like finger-gaiting and in-hand reorientation [56, 57]. In Section 6, we evaluate the ability of the representations to track SE(2) pose changes of the object relative to the sensor [T3] [10], prediction of the stability of a grasp [T4] [8], and textile recognition [T5] [58].
**3. Do the representations enable manipulation planning?** Pre-trained representations can provide tactile features to a manipulation policy, improving training efficiency and test performance by eliminating the need to extract the states from raw sensor data. In Section 7, we design a bead maze [T6] manipulation problem as illustrated in Figure 3 (c), where the robot using tactile sensing is tasked to move a bead along a curved wire.

**Evaluation protocol.** We adopt a frozen evaluation procedure with an encoder-decoder architecture. Specifically, we *freeze* the pre-trained Sparsh encoder weights and train the parameters of an attentive decoder [51, 59] to assess what touch representations have captured from self-supervised pre-training alone. All tasks in the benchmark, except force field visualization and policy learning, train an

attentive decoder containing a cross-attention module and a two-layer MLP using labeled datasets from Table 3. We also include an end-to-end (E2E) baseline with identical model capacity where the same encoder and decoder probe are initialized with random weights and all the parameters (both encoder and decoder) are trained. Further, we train downstream decoders with different amounts of labeled data to evaluate task performance under progressively low labeled data regimes.

In the following sections, we describe the design, metrics and results of each task in TacBench. Additional details are provided in Appendix D, and ablations with unfrozen Sparsh encoder, encoder model sizes, and few-shot cross-sensor transfer are provided in Appendix E.

## 5 Comprehending tactile properties

### 5.1 [T1] Force estimation

**Task.** Force estimation is defined as the prediction of 3-axis normal and shear forces applied on the sensor's elastomer. Figure 3(a) shows our data collection setup. We use three different indenter shapes to collect force-labeled data: hemisphere, sharp, and flat. Our dataset contains 75k time-aligned samples of 3-axis force measurements, end-effector poses, and tactile images from DIGIT at 60fps and GelSight at 25fps. We train the decoder using normalized force measurements scaled between $[-1, 1]$, supervised using $\mathbf{L_1}$ loss and optimized using Adam until convergence. We compare performance using the average root mean squared error (RMSE) across all three axes.

**Results.** GelSight Mini images are of higher resolution (HD) compared to DIGIT ($320 \times 240$) resulting in smaller contact regions against the background. For this reason, we observe that when sufficient supervised data is available for DIGIT, it is possible to train a model from scratch to achieve high accuracy, but for GelSight Mini the end-to-end model does not perform well. Figure 4 (i)-(ii) shows across the board that our frozen Sparsh representations can estimate forces with low error. Specifically, we find Sparsh (DINO) to be robust even when access to labeled data is sparse, a common scenario in tactile sensing. Additional details are in Appendix D.3.

### 5.2 [T1A] Force field visualization

We qualitatively evaluate the representations for rendering normal and shear force fields to understand sensor-object interactions. Although obtaining a shear field for sensors with markers nowadays is trivial via marker tracking [54], it is challenging and underexplored for markerless sensors. We train a CNN decoder using the reassemble-fusion approach for dense predictions [60] unsupervised, since we do not have access to ground truth for markerless sensors. We frame normal field estimation as depth estimation [61] and shear field estimation as optical flow [62, 63, 64, 65]. Figure 4 (vi) shows visualizations for the top-performing model Sparsh (DINO) in [T1] that provides directional information about the relative motion of the contact patch. For instance, sliding motion (a, c, e, f), torsional slip (b), and divergence field upon contact (d). Additional details are in Appendix D.4.

### 5.3 [T2] Slip detection

**Task.** Shear and slip are closely related. Using the same setup as force estimation, we collect strokes where a hemispherical probe slides over the sensor, producing trajectories with both sticking and slipping samples. Slip is labeled using the friction cone model with an empirically estimated static friction coefficient (see Appendix D.5). The dataset, with a notable imbalance between no-slip and slip classes, contains 125k samples with 13% slip instances. We train two decoders: one for slip detection and another for normalized force changes ($\Delta$) as we find that predicting the two correlated quantities jointly enhances slip detection. The MLP decoders use cross-entropy for slip detection and mean absolute error for $\Delta$ force regression, reserving 25k samples for evaluation.

**Results.** We report F1 score instead of accuracy due to the imbalance in the slip labels in the dataset. Figure 4 (iii)-(iv) illustrates the advantages of frozen Sparsh features trained under a JEPA paradigm for slip detection, particularly challenging for DIGIT sensor, even when using only 1% of the training dataset. In particular, Sparsh (VJEPA) achieves the highest F1 score among the models. Although all models detect slip from the 80 ms history of tactile data, Sparsh (VJEPA) benefits from a detailed temporal perspective, as its encoder processes a video clip with four frames spanning this window. Sparsh backbones also show better performance than the E2E model when labeled training data is significantly reduced. Additional details are in Appendix D.5.

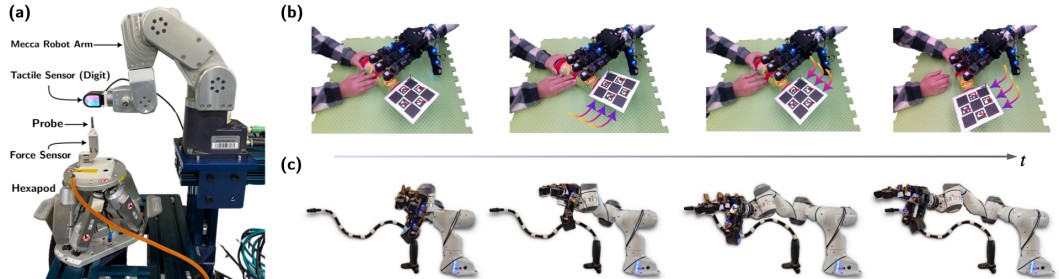

**Figure 3:** Real labeled data collection setup for `TacBench` tasks (a) `[T1]` Force estimation and `[T2]` Slip detection, (b) `[T3]` Pose estimation, and (c) `[T6]` Bead maze.

## 6 Enabling physical perception

### 6.1 `[T3]` Pose estimation

**Task.** Estimating object pose changes can help tasks such as tracking object drift for in-hand translation [66], rotation [56, 57], and pushing [7], among others. Given that tactile images capture local changes between sensor-object, we evaluate `Sparsh` representations to estimate $\mathbf{SE}(2)$ transformations of the object relative to the sensor. Figure 3 (b) illustrates the data collection procedure. The dataset consists of time-synchronized pairs of DIGIT observations $\mathbf{z}_t \in \mathbb{R}^{h \times w \times 3}$ and object poses $\mathbf{T}_t \in \mathbf{SE}(3)$. $\mathbf{T}_t$ are then preprocessed to produce relative pose changes on the sensor gel as $\mathbf{S}_t^{t-1} \triangleq (\Delta x, \Delta y, \Delta \theta) \in \mathbf{SE}(2)$. We follow the regression-by-classification paradigm for this task [10, 66]. Relative object poses are binned into a grid, capturing translations with a resolution of $\pm 5$mm and rotations with a resolution of $\pm 2°$. For each degree-of-freedom (DOF), we train a head to predict probability distribution over the discretized grid using cross-entropy loss and Adam optimizer.

**Results.** Multiclass accuracy reveals that E2E approaches perform well with ample data, but drastically decline when labeled data is reduced, as shown in Figure 4 (v). Small datasets make it difficult to distinguish between close categories, such as orientation changes from $[0.5°, 1.0°]$ to $[1.0°, 2.0°]$. Pre-trained representations, however, maintain good performance even with only a third of the data. In low data scenarios, decoders using tactile representations often revert to extreme values, reducing estimation resolution and accuracy. Additional details and examples are provided in Appendix D.6.

### 6.2 `[T4]` Grasp stability

**Task.** Grasp stability is well-studied in the tactile sensing literature for parallel jaw grippers [40, 67, 68, 69]. We evaluate whether representations aid in predicting grasp success given a short history of tactile images from a single finger. Specifically, we take inspiration from [8] and adapt the Feeling of Success dataset. Each sample consists of a triplet of tactile images corresponding to 'before', 'during', and 'after' grasping a set of objects. The dataset consist of $64\%$ successful grasps and $36\%$ failed grasps. We pass to the SSL model the 'before' and 'during' as tactile history. Since [8] does not specify an official train/test split, we create our randomized split with all objects, using approximately 8k grasps for training and the remaining 1.3k grasps for evaluation.

**Results.** Training with the full dataset, all models achieve similar accuracy. `Sparsh` (IJEPA) or `Sparsh` (VJEPA) reach $\sim 80\%$ classification accuracy, surpassing results from [8] that combined tactile and vision modalities as shown in Figure 4 (vii). Our model, relying solely on touch from a single finger, shows competitive performance even with only 33% and 10% of the data. However, with just 80 training samples, performance drops significantly. More details are in Appendix D.7.

### 6.3 `[T5]` Textile recognition

**Task.** Vision-based tactile sensors are broadly used for material property recognition, since their compliant gel and high resolution cameras make them effective at discriminating different materials by surface texture [58, 39]. Specifically, we take the task definition from [58] and adapt the Clothing Dataset. The dataset consist of 4467 short video clips (10-25 frames), of a robot with a GelSight 2017 (markers) grasping several types of textile (20 classes), such as leather, cotton, polyester, etc. We follow the train-test split provided in the metadata of the dataset.

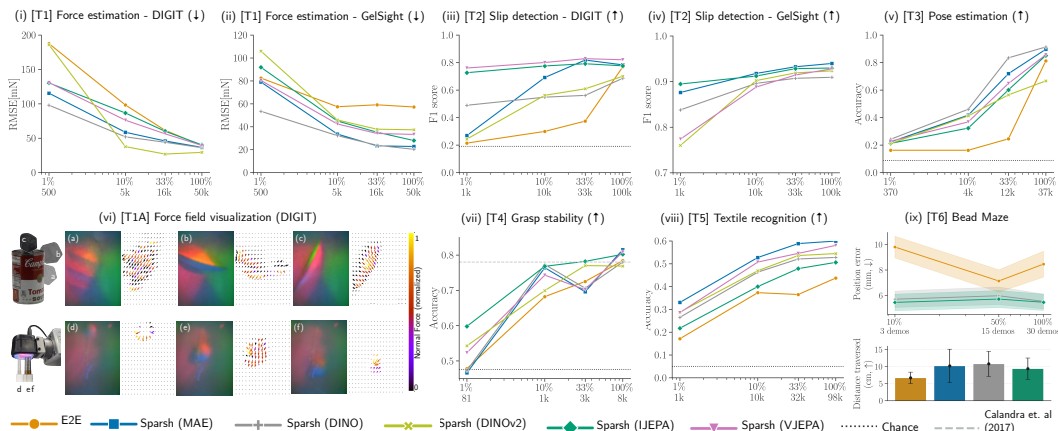

**Figure 4:** Summary of results comparing `Sparsh` and `E2E` on [T1]-[T6] tasks in `TacBench` across varying amounts of labeled data. Pre-training with SSL yields general touch representations that work across several tasks and sensors outperforming task and sensor specific models particularly under limited labeled data budget.

**Results.** Training an `E2E` specialist model for textile recognition using the full dataset can be challenging, as noted in [58]. By leveraging pre-trained touch representations, as shown in Figure 4 (viii) the performance of the task can be significantly improved, even when training with only 10% of the labeled data. `Sparsh` (`MAE`) is particularly effective, as it heavily relies on pixel-level features (see Appendix D.8). Additionally, we evaluate the cross-sensory ability of the representations, finding that with few samples (10-shot) `Sparsh` quickly adapts the downstream task to DIGIT (see Appendix E.3).

## 7 Enabling manipulation planning

### 7.1 [T6] Bead maze

**Task.** The bead maze is a children's toy to enhance fine motor skills. We adapt this task to robot policy learning, where the goal is to guide a bead from one end to another following the path of the wire (maze). Given a small history of tactile images $(\dots, \mathbf{z}_{t-1}, \mathbf{z}_t)$, and robot proprioception $(\dots, q_{t-1}, q_t)$, we train a policy to predict changes in joint angles as actions $\mathbf{a} \triangleq (\Delta q_t, \Delta q_{t+1}, \dots); \Delta q \in \mathbb{R}^7$, to make progress on this task. This task is fundamentally tactile-focused, as the robot needs to react to resistance encountered by changes in the maze pattern and the subtle local movement of the bead are difficult to perceive from vision even when not occluded by the hand. A prior version of the bead maze task has been explored in robotics relying solely on tactile feedback [70]. In our setup, illustrated in Figure 3 (c), we assume that the robot starts with an initial stable grasp. We collect a dataset of 50 demonstrations on different maze patterns with a mix of VR-based and manual kinesthetic-based teleoperation, corresponding to a total of ∼34k training pairs of tactile images and robot joint angles. Since we are training policies with real data, we use diffusion policy [71] for this task as it is one of the leading behavior cloning methods. For tactile observation conditioning, we replace the vision encoder in Diffusion Policy with the pre-trained `Sparsh` encoder.

**Results.** We evaluate `Sparsh` (`DINO`) and `Sparsh` (`IJEPA`) for policy learning, as these representations exhibit the best performance across the rest of the benchmark. For completeness, we also consider `Sparsh` (`MAE`), and `E2E` which trains a tactile encoder and policy end-to-end. Due to covariate shift [72] in behavior cloning, prediction errors can accumulate over time; therefore, we report position error between the predicted trajectory and a demonstration trajectory from a held-out maze sequence over small chunks of 3cm followed by the robot corresponding to 15 timesteps of action predictions. Figure 4 (ix) shows the position error over access to different number of demonstrations for training. We find that `Sparsh` (`DINO`) and `Sparsh` (`IJEPA`) produce significantly (a difference of ∼16%) lower trajectory errors compared to training the policy `E2E`.

Additionally, we evaluate real rollouts of the learned policies (using all 50 demonstrations) over a set of 10 randomized novel starting locations on the maze. In Figure 4 (ix) we report distance traversed (in cm) before failure. We find that policies using `Sparsh` representations outperform `E2E` by ∼20-53%. We note that given the high precision nature of this task and the considerations for real system deployment for the policy, none of the models succeeds in completing the full maze on real

robot rollouts. We expect that increasing the diversity of training data with different maze patterns will highlight the generalist capabilities of touch representations, and that temporal ensembling will aid in improving the smoothness of the policy [73]. Additional details are in Appendix D.9.

# 8 Discussion

**Summary.** We present Sparsh, a family of general touch representations trained with self-supervision for vision-based tactile sensors. We learn general-purpose, cross-sensor representations from a curated, unlabeled dataset of 460k+ samples from DIGIT, GelSight 2017, and GelSight Mini sensors. We evaluated five SSL approaches (see Figure 2) comparing their performance against task and sensor specific models through TacBench, a benchmark of six touch-centric tasks designed to assess the content and quality of the representations. Our results indicate that Sparsh representations are performant across various sensors and tasks capturing tactile properties, and enhancing physical perception and manipulation planning.

**Analysis.** Overall Sparsh excels on all tasks. In particular, we find Sparsh (DINO) is well suited for physics-based tasks like force and pose estimation, while Sparsh (IJEPA) performs better at touch semantic understanding like slip state, stability of a grasp, and textile recognition. On average Sparsh (DINO) outperforms Sparsh (IJEPA) by $5.6\%$ across the benchmark. Both models perform similarly in bead maze test demonstrations, which require implicit knowledge of shear forces and slip. However, this did not translate to real robot performance due to lack of force control and system-level confounding variables not captured during training. These include the high precision required to keep the bead in place, the impossibility of error recovery once grip is lost, and trajectory drift due to local decision-making. Specialist policies or models trained from scratch exhibiting better robot rollout performance is due to the narrow task domain setting that leads to overfitting, a trend similarly observed when studying pre-trained vision models [71, 73, 74].

Learning touch representations in latent space is more advantageous than in pixel space, as these representations can filter out and generalize over noise or lighting differences. Tasks traditionally challenging for markerless sensors (like DIGIT and GelSight Mini), such as shear force (and field) estimation and slip detection, become solvable with our general touch representations. On average, Sparsh achieves a $95.1\%$ improvement compared to an end-to-end approach when all models have access to only $33\% - 50\%$ of the labeled dataset per downstream task. Using as little as $10\%$ or $1\%$ of the labeled data for force estimation and slip detection still yields acceptable results (e.g. force error below 0.1N with Sparsh (DINO)). Fine-tuning Sparsh encoders is another method of assessing the quality of pre-trained representations. We provide in Appendix E.1 experiments with partial and full fine-tuning. Notably, models pre-trained in latent space perform better in downstream tasks when fully fine-tuned, especially in regression tasks like force and pose estimation. In contrast, partial fine-tuning offers minor improvements, aligning closely with the performance of frozen models. We also evaluate Sparsh decreasing the model capacity, finding the biggest impact in performance for regression-like tasks when training with limited amount of labeled data (see Appendix E.2).

Sparsh is a significant step towards a general pre-trained backbone for vision-based tactile sensors. Our aim is to enable efforts to compile larger tactile datasets that include additional vision-based tactile sensors and leverage the benefits of scaling up SSL backbones, as seen in computer vision and natural language processing. TacBench serves as an initial benchmark for evaluating these representations, and additional tasks can be incorporated based on the needs of the tactile sensing and manipulation community. For instance, further exploration of pre-trained touch representations in tactile policy learning, or tracking dynamic object properties like changing mass during pouring.

**Limitations.** Open-source tactile datasets we considered in this study predominantly feature discrete contact interactions. We believe that incorporating data rich in shear interactions can further improve the representations. We do not ablate the length of tactile image history for learning the representations. Such ablations could provide guidance on improving their quality for downstream tasks. Our bead maze policies with pre-trained touch representations deployed on the real robot are only able to complete the maze partially before compounding error leads to the bead falling out of the fingers. Further research is needed to understand how to effectively leverage pre-trained touch representations in behavioral cloning for robot manipulation tasks.

**Acknowledgments**

The authors thank Ishan Misra, Mahmoud Assran for insightful discussions on SSL for vision that informed this work, Changhao Wang, Dhruv Batra, Jitendra Malik, Luis Pineda, Tess Hellebrekers for helpful discussions on the research, and Ishan Misra, Homanga Bharadhwaj, Tarasha Khurana, Rosario Scalise for feedback on the paper draft.

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

# Appendix

## A  Contributions

The contributions of the authors are as follows.

**Carolina Higuera** led the execution of the project, designed the experiments, the data collection protocol, implemented some of the SSL pretraining methods and downstream tasks, did bug fixes, code reviews, performance evaluations, and wrote the paper.

**Akash Sharma** led the execution of the project, was involved in experiment design, implemented core functionality including SSL pretraining, conducted performance evaluations and benchmarking, did bug fixes, code reviews, and wrote the paper.

**Krishna Bodduluri** collected and curated labeled data for downstream tasks such as force estimation and slip classification.

**Taosha Fan** implemented SSL pretraining and evaluation infra, did bug fixes, code reviews, advised on the evaluations, provided feedback and helped revise the paper.

**Patrick Lancaster** helped with hardware setup for the bead maze task, provided feedback and helped revise the paper.

**Mrinal Kalakrishnan** advised on the project, supported the research team.

**Michael Kaess** advised A.S., provided feedback on evaluations and the paper.

**Byron Boots** advised C.H., provided feedback on evaluations and the paper.

**Mike Lambeta** provided insights on tactile sensing and informed model input design choices, provided feedback on experiment design and evaluations.

**Tingfan Wu** helped design data collection protocols, steered project engineering, provided feedback on experiment design and evaluations, and helped revise the paper.

**Mustafa Mukadam** led the project, set the vision and research direction, steered the team and provided guidance on all aspects of the project from approach to evaluations, and helped write the paper.

## B  Broader related work

**Self Supervised Learning.** We detail recent developments in masking-based self-supervised learning approaches.

*Masked Image Modeling* (MIM) is the strategy of corrupting a data sample by significantly masking a portion of the sample and training a model to recover the missing portion, conditioned on the corrupt sample. It has become a prominent framework in SSL with the success of [23, 75]. An important design consideration here is the output space of the model for supervision, which can be either raw pixels [23, 76] or an alternative representation space [77, 78, 79, 75]. While training Masked auto-encoders is simple, these models are comparatively sample inefficient during training [50].

*Self-distillation* [80] is the idea of training two (usually identical) networks such that a *student* network learns to predict the output representations of a *teacher* [81] network via a small predictor network when observing augmentations of the same data sample. It has been shown to improve performance significantly even in the case of abundant data [82]. While degenerate constant representations is a concern, a common strategy is to stop gradient backpropagation [25] through the teacher network and employ momentum based weight updates [22]. A concrete instance is DINO [48] utilizing ViTs [83] as the student & teacher encoder networks. More recently DINOv2 [49] improved downstream performance significantly by combining self-distillation and MIM.

*Joint-Embedding Predictive Architectures* (JEPA) [52] share similarities with MIM, as both rely on masking. However, the JEPA framework conceptually prescribes two key changes: a) information restoration in a latent representation space, rather than in input space (pixels or tokens) b) prediction of latent embedding conditioned on the *masking parameters*. This framework has had success across various modalities, including audio [84, 85], images [50, 86], and pointclouds [87]. Notably, in this paper we consider masking strategies from I-JEPA [50] and V-JEPA [51]. I-JEPA utilizes a spatial block-masking strategy and V-JEPA utilizes tube-masking [88] with varying aspect ratios for learning representations efficiently in latent space circumventing decoding unnecessary pixel-level details.

**Representation learning in robotics.** Pretraining models for multi-task capability has become popular recently, especially after the success of self-supervised learning (SSL) in computer vision tasks like object classification, segmentation, depth estimation, and image generation. These tasks, while typically tested on computer vision datasets, are also very common in robotics. The idea of using these pre-trained representations for robot learning was initially explored in [89], showing that pre-trained visual representations can sometimes even be better than using ground-truth state representations for training control policies.

Generative SSL via masked image modeling (MIM) [90, 91] has shown successful transfer of pre-trained representations from in-the-wild data to real-robot scenarios, enabling basic motor skills such as reaching, pushing, and picking. Furthermore, many other works investigate contrastive learning approaches to learning general visual representations in robotics [92, 93, 44]. These methods usually employ a pixel reconstruction objective based on a time-contrastive objective or focus on contrasting video clips leveraging natural language for video-language alignment.

The field has been moving towards finding general-purpose representations that work well across a wide range of problems in robot manipulation learning. Voltron [94], is a framework for language-driven visual representation learning for robotics that combines both masked auto-encoding and contrastive learning techniques, focusing on multi-task performance. This model is trained to learn representations that capture both low-level spatial reasoning and high-level semantic understanding by using language supervision from human videos.

**Tactile sensor simulation.** Multiple simulators have been proposed for vision-based tactile sensors such as [95, 96, 97, 98, 99] with the hope of sim2real generalization of learned policies [100]. However, many of these methods are either limited to marker-based tactile sensors [100], or narrow tasks [101, 102]. Certain other methods [40] also leverage simulated data to train multi-modal representations. However, in general we find that tactile simulators are still unable to model shadows, as well as real-world per-sensor-instance discrepancies, hampering their potential use for representation learning.

## C   Touch representation and self-supervision details

To ensure fair evaluation of all models, our SSL algorithms are largely adapted from official MAE, I-JEPA, V-JEPA, DINO, DINOv2 codebases.

### C.1   Training details

We train all models on 8 Nvidia A-100 (80G) GPUs. In addition to training losses, to monitor training progress, we rely on online probes. Specifically, we find that for joint embedding predictive architectures, the training losses are not indicative of model convergence during optimization; therefore, proxy metrics such as reconstruction quality are helpful. For all methods, we utilize DPT [103] based decoders to decode the tactile representations back into tactile images. See Figure 5 for some examples of tactile reconstructions from Sparsh embeddings. All encoder models are trained for 150 epochs. We use AdamW optimizer and use a linear rampup followed by a cosine schedule as the learning scheduler. Further, we find that tuning momentum value as well as the weight decay factor was important in observing training convergence without collapse. Additional information of hyperparameters is detailed in Table 1.

|                | Arch.    | EMA decay | LR      | Batch size |
|----------------|----------|-----------|---------|------------|
| Sparsh (MAE)   | ViT-B/14 | N/A       | 1e-4    | 100        |
| Sparsh (DINO)  | ViT-B/14 | 0.998     | 1e-4    | 150        |
| Sparsh (IJEPA) | ViT-B/14 | 0.996     | 6.25e-4 | 150        |
| Sparsh (VJEPA) | ViT-B/14 | 0.996     | 6.25e-4 | 150        |

**Table 1: Training hyperparameters for Sparsh models.** All models run for 150 epochs with optimizer AdamW, a weight decay cosine schedule from 0.04 to 0.4, and a learning rate warmup of 30 epochs.).

### C.2   Architecture details

All encoder models are Vision Transformers (ViT) [83]. Although the main encoder models use ViT-B/14 as the standard architecture, following [50] we use a small ViT as the predictor network.

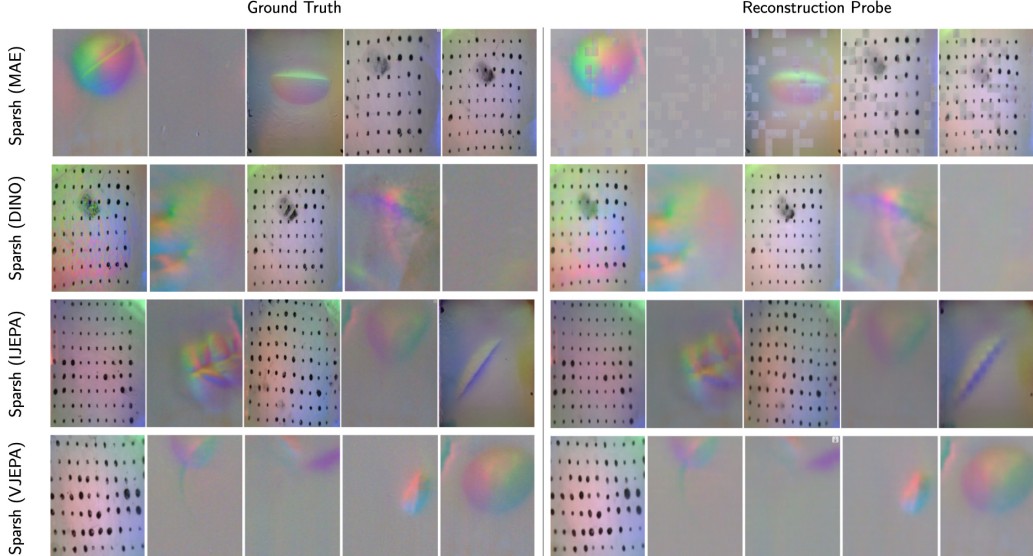

**Figure 5:** Visualization of reconstructed tactile images using the online probe to monitor SSL training of `Sparsh` models.

|  | Sparsh (MAE) | Sparsh (DINO) | Sparsh (IJEPA) | Sparsh (VJEPA) |
|---|---|---|---|---|
| N. parameters | 86254848 | 86255616 | 86386944 | 86537472 |
| FPS | 104 | 112 | 112 | 60 |

**Table 2:** Number of parameters and inference time for `Sparsh` backbones

All the models are pretrained without a `[cls]` token. For DINO, which decodes the `[cls]` token into classes, we repurpose ViT registers [104] to predict classes. In Table 2 we report the number of parameters for each encoder and their respective inference times.

Tactile images with a stride of 5 i.e., $\mathbf{I}_t \oplus \mathbf{I}_{t-5} \in \mathbb{R}^{h \times w \times 6}$ are concatenated along the channel dimension before the background is removed and reshaped to $224 \times 224$ for ViT processing. We choose a stride of 5 as consecutive images are similar due to high sensor sampling rates, and to match the slip detection window in humans. Ablating the effect of the input image and patch resolution may be important for better performance and is left for future work.

### C.3 Dataset splits

We use three available datasets for training `Sparsh`, namely YCB-Slide [9], Touch-and-Go [20] and Object Folder [37]. The YCB-Slide dataset consist of human sliding interactions with 10 YCB objects. Each object has 5 trajectories, with around 3500 frames each from DIGIT sensors with different optical characteristics (180k frames in total). For each object, we dedicate four trajectories for training and the last one for validation. Touch-and-Go consists of discrete human contact interactions with in-the-wild objects, using a GelSight sensor. It consist of 140 videoclips and plain files with labels for the frames with a clear contact. We use all frames (220k) in the videoclips since we do not rely on labeled data for SSL training, from which 70% is used for training and the remaining for validation. The data used from ObjectFolder consist of 81k frames of robot discrete contact interactions with objects in a controlled setting. We also use a train/val split of 70/30.

To complement the dataset, we collected Touch-Slide with additional human sliding interactions on toy-kitchen objects with the DIGIT sensor. We use 9 objects, shown in Figure 6 and collected 5 trajectories for each, generating 180k frames in total.

For all downstream tasks we use tactile data from real sensors/hardware (DIGIT, GelSight17, and GelSight Mini) that were not seen during `Sparsh` SSL training. Under our problem formulation, this allows us to investigate generalization of `Sparsh` to new sensor instances (consider the case of swapping out a sensor from a robotic hand due to wear-off).

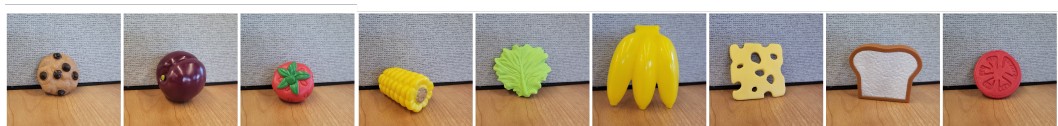

**Figure 6:** Set of objects for collecting sliding contact trajectories in the Touch-Slide dataset.

Similarly, all objects used for downstream tasks were not used for SSL training. For example, [T1] and [T2] tasks use a real robot arm to slide the sensor elastomer (DIGIT, GelSight Mini) against an indenter to collect force-labeled data. [T4] uses an open source dataset for grasp stability [4], which includes data from a real robot grasping over 100 unique objects using an unseen GelSight17 with printed markers during SSL training. Similarly, [T6] uses DIGIT data collected from real hardware, a robot pulling and moving a bead along the wire. Note that none of the data used for learning representations comes from this kind of object-robot hand interactions.

### C.4  Short summary of SSL methods

In this paper, we consider three SSL paradigms, namely Sparsh (MAE), Sparsh (DINO), and Sparsh (IJEPA) & Sparsh (VJEPA).

**Sparsh (MAE)** is based on the principle of masked image modeling, where an encoder model is tasked with learning the contextual representations of substantially masked images, such that it enables reconstruction of the masked regions via a lightweight decoder. We use a ViT encoder and decoder for Sparsh, and the MAE loss corresponds simply to a L2 reconstruction loss:

$$\mathcal{L}_{\text{MAE}} = \|\mathbf{I}_{\text{target}} - \mathbf{I}_{\text{recon}}\|_2^2 \tag{1}$$

**Sparsh (DINO)** is based on the principle of self-distillation between two identical networks, where a student network learns to track the output predictions of a EMA teacher network. Cross-entropy loss is employed between the predictions of the student and teacher network, both of which consume different crops of the same input. Specifically, feature representations from each branch are passed through a MLP head, producing probability vector over an arbitrarily chosen number of classes. These scores are normalized to produce $\mathbf{p}_s$ and $\mathbf{p}_t$ for the student and teacher respectively.

$$\mathcal{L}_{\text{DiNO}} = -\sum \mathbf{p}_t \log \mathbf{p}_s \tag{2}$$

**Sparsh (IJEPA) and Sparsh (VJEPA)** share similarities with both masked image modeling and self-distillation. Here, we employ two identical networks termed context and target networks. The context network corresponds to a student network, which is tasked to predict the features from a EMA target network (teacher network), through a small predictor network. In this case, a $L_2$ loss over features is used to enforce similarity between the two branches. Specifically, the context network observes $M$ global masks of an image to produce contextual features, which are then passed through a predictor to predict target network features of $B_i$ local crops of the same image $\hat{\mathbf{s}}_{y_j}$. On the other hand, the target network consumes local crops of the image to produce $\mathbf{s}_{y_j}$.

$$\mathcal{L}_{\text{jepa}} = \sum_{i \in M} \sum_{j \in B_i} \|\hat{\mathbf{s}}_{y_j} - \mathbf{s}_{y_j}\|_2^2 \tag{3}$$

## D  TacBench tasks and evaluation details

### D.1  Labeled datasets

See Table 3 for details on labeled data curation for TacBench tasks.

### D.2  Probe details

The parameters of the model updated via EMA (target encoder for Sparsh (IJEPA) and Sparsh (VJEPA), teacher network for Sparsh (DINO) and Sparsh (DINOv2), encoder from Sparsh (MAE)) are fixed and used for evaluation. The features are pooled via attentive pooling for tasks that require global representations, such as slip detection, resultant force estimation, and classification tasks. For tasks that require dense reasoning, we use DPT decoders [103] to decode patch representations into full input resolution quantities such as normal and shear force fields, and reconstructed tactile images. See Figure 7 for a visual illustration of the probe architectures.

| Task | Dataset | Sensor | Size | Collector | Label |
|------|---------|--------|------|-----------|-------|
| [T1] Force estimation | Shear load (indenter: sphere, flat, sharp) | DIGIT | 75k | Robot | 3-axis force |
| | | GelSight Mini | 75k | Robot | 3-axis force |
| [T2] Slip detection | Shear load (indenter: sphere) | DIGIT | 125k | Robot | Friction cone |
| [T3] Pose estimation | Object sliding | DIGIT | 49k | Human | Object pose $\mathbf{SE}(2)$ |
| [T4] Grasp stability | Feeling of Success [8] | GelSight 2017 | 9.3k | Robot | Success (yes/no) |
| [T5] Textile recognition | Clothing Dataset [58] | GelSight 2017 | 120k | Robot | Textile ID |
| [T6] Bead maze | Demonstrations | DIGIT | 34k | Human | Joint angles |

**Table 3:** Datasets in TacBench for evaluating representations on downstream tasks.

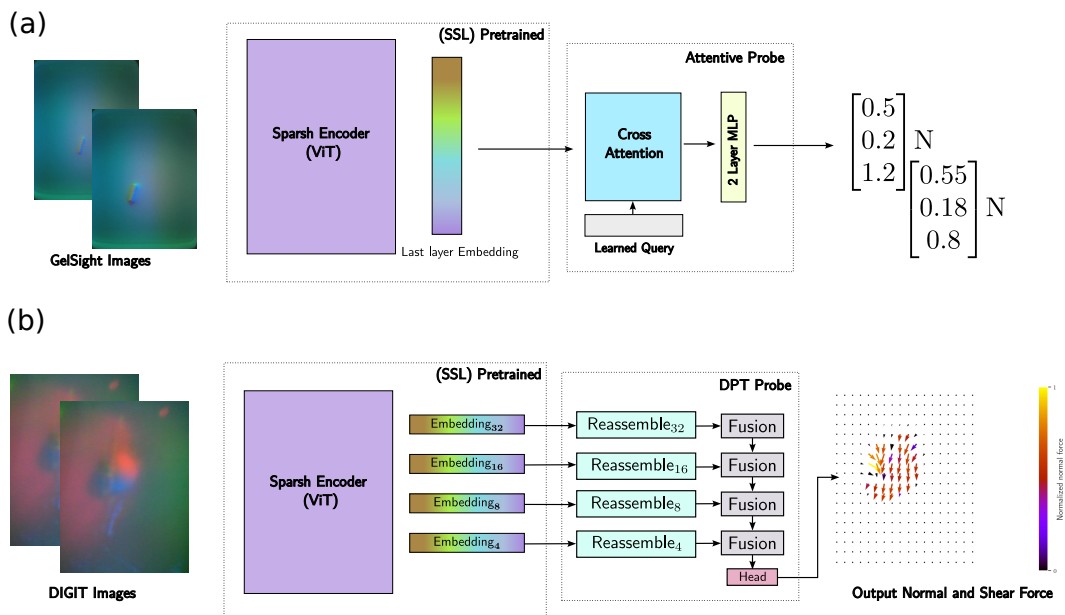

**Figure 7:** (a) Attentive probe architecture consists of a cross attention layer followed by a linear layer to regress *resultant* output quantities such as resultant force or slip state (b) Dense Prediction Transformer (DPT) [103] consists of multiple reassemble and fusion layers to decode features from intermediate layers of the Sparsh backbones to produce dense outputs such as normal and shear fields

We follow attentive probing[51, 59] to assess the capabilities of tactile representations on the benchmark, as this approach allows us to determine what representations capture from self-supervision alone. For most tasks – except force field visualization and policy learning – in the benchmark, we freeze Sparsh and train a cross-attention module (hyperparameters in Table 4) followed by a light 2-layer MLP probe supervised, using the labeled dataset for each task.

| Parameter | Setting |
|-----------|---------|
| Embedding dimension | 768 |
| N heads | 12 |
| MLP ratio | 4.0 |
| Depth | 1 |
| Layer normalization | Yes |

**Table 4:** Attentive pooling hyperparameters used for evaluation protocol of representation in downstream tasks.

### D.3 [T1] Force estimation

After attentive pooling, the tactile features with 768 dimensions are passed to a 2-layer MLP with 192 and 3 units respectively, to get the 3-axis force estimations. Two independent force decoders are trained using DIGIT and GelSight Mini data respectively, using the sharp and sphere probe data during training and the flat indenter data for testing. The target forces are normalized to be $\pm 1.0$ and scaled back after prediction. We train the force decoder using Adam optimizer with 1e-4 learning rate.

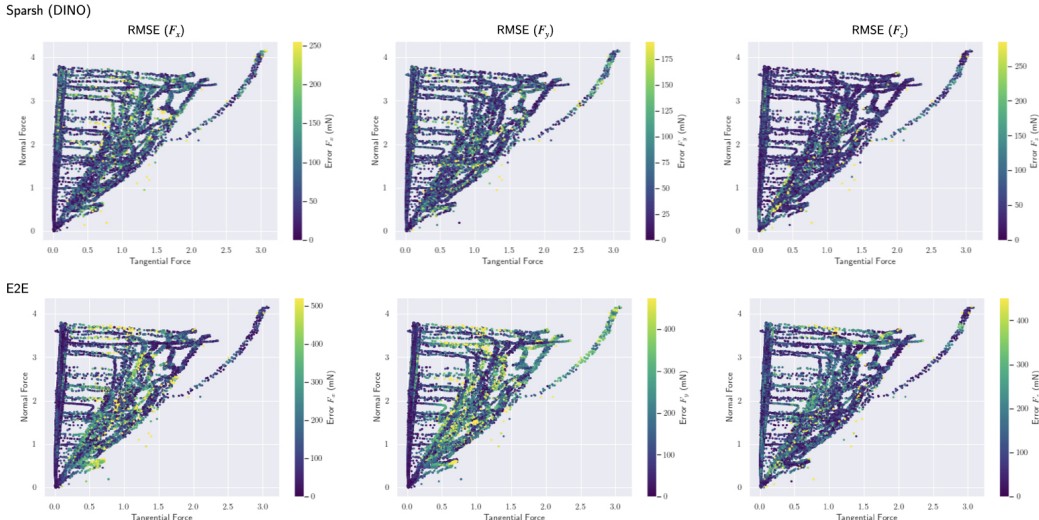

**Figure 8:** Friction cone of test data and RMSE (mN) for force estimation task with DIGIT sensor.

**DIGIT.** In Table 5 we report the average RMSE over 25k samples of unseen DIGIT data for the force estimation task. We report metrics for each Sparsh model and the E2E approach, under four different budgets of training data. We also provide a 95% confidence interval to ground the error ranges of each model.

In Figure 8 we plot the friction cone from the test data, where the colormap represents the error in mN for each axis. Note that E2E exhibit larger errors (around 500mN) for the tangential component and they are more predominant as the normal force increases. In contrast, the top model Sparsh (DINOv2) estimates with low error ($< 100$mN) in general across the whole range of tangential and normal forces.

| Model | Full dataset (50k) | 1/3 dataset | 1/10 dataset | 1/100 dataset |
|---|---|---|---|---|
| E2E | 39.34 [39.21, 39.48] | 61.42 [61.12, 61.72] | 98.22 [97.61, 98.84] | 187.51 [185.51, 188.51] |
| Sparsh (MAE) | 36.61 [36.51, 36.71] | 45.96 [45.80, 46.12] | 58.55 [58.31, 58.79] | 115.39 [114.69, 116.09] |
| Sparsh (DINO) | 36.09 [36.01, 36.17] | 44.03 [43.87, 44.19] | 51.89 [51.69, 52.10] | **97.95** **[97.36, 98.52]** |
| **Sparsh (DINOv2)** | **29.31** **[29.14, 29.46]** | **26.85** **[26.70, 26.99]** | **37.66** **[37.45, 37.86]** | 185.86 [184.94, 186.78] |
| Sparsh (IJEPA) | 40.27 [40.16, 40.38] | 60.04 [59.72, 60.34] | 86.57 [86.06, 87.08] | 130.37 [129.59, 131.15] |
| Sparsh (VJEPA) | 39.38 [39.30, 39.47] | 56.34 [56.07, 56.62] | 76.11 [75.67, 76.55] | 130.83 [130.29, 131.38] |

**Table 5:** Root Mean Squared Error (mN) and 95% confidence interval for force estimation with DIGIT data. All models were evaluated on flat indenter data over 25k test samples.

**GelSight.** In Table 6 we report the average RMSE over 25k samples of unseen GelSight data and the corresponding 95% confidence interval. Notice from Figure 9 that the majority of errors are localized around the dynamic shear region. It is worth noting that the errors associated with Sparsh (DINO) remain below 150mN, whereas E2E exhibits higher errors, particularly in the estimation of normal forces.

### D.4 [T1A] Force field visualization

Since rendering the force field is a dense prediction task, we do not apply the attentive probing protocol. Instead, we follow DPT [60], training a CNN encoder with reassemble-fusion modules at layers 2,5,8,11 of the Sparsh encoder to progressively upsample the representations to obtain

| Model | Full dataset | 1/3 dataset | 1/10 dataset | 1/100 dataset |
|---|---|---|---|---|
| E2E | 57.21 [56.44, 57.98] | 59.09 [58.15, 60.04] | 57.43 [56.44, 58.42] | 82.42 [80.98, 83.86] |
| Sparsh (MAE) | 22.72 [22.27, 23.17] | **23.28** [22.83, 23.72] | 33.56 [33.04, 34.08] | 78.98 [77.74, 80.21] |
| **Sparsh (DINO)** | **20.25** [19.85, 20.65] | 23.79 [23.40, 24.18] | **32.17** [31.67, 32.67] | **53.43** [52.69, 54.17] |
| Sparsh (DINOv2) | 37.30 [36.71, 37.88] | 37.79 [37.22, 38.37] | 45.86 [45.14, 46.59] | 105.95 [104.28, 107.62] |
| Sparsh (IJEPA) | 27.91 [27.37, 28.44] | 35.20 [24.57, 35.82] | 44.93 [44.13, 45.73] | 91.81 [90.76, 92.86] |
| Sparsh (VJEPA) | 33.26 [32.67, 33.84] | 34.07 [33.39, 34.75] | 42.35 [41.60, 43.10] | 80.36 [79.26, 81.47] |

**Table 6:** Root Mean Squared Error (mN) and 95% confidence interval for force estimation with GelSight Mini data. All models were evaluated on flat indenter data over 25k test samples.

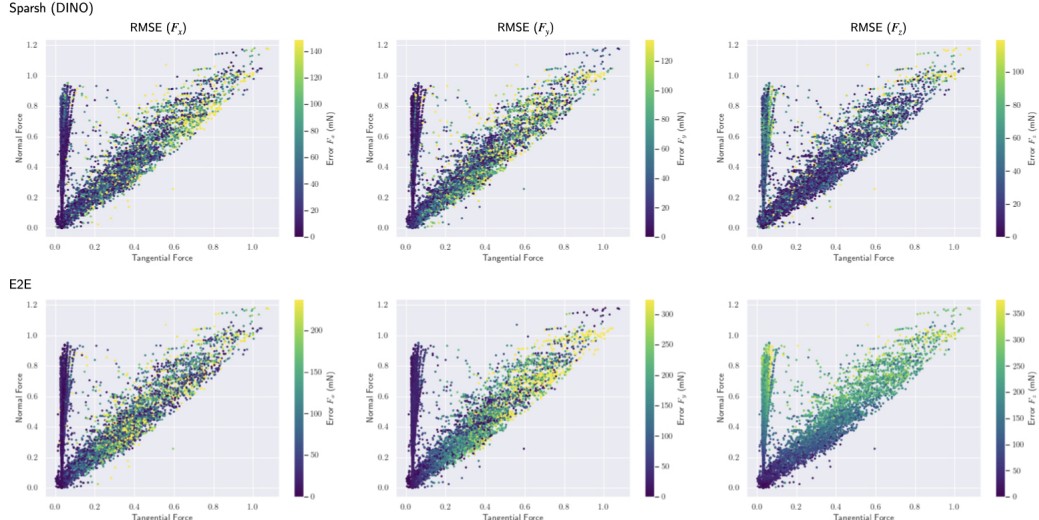

**Figure 9:** Friction cone of test data and RMSE (mN) for force estimation task with GelSight sensor.

a fine-grained prediction of the force field. After the reassemble-fusion modules, we attach two task-specific task head, for normal and shear field prediction.

Since for markerless vision-based sensors it is not trivial to get ground truth of the force field, we turn to unsupervised learning. Depth estimation and optical flow are analogous to the estimation of normal and shear force fields, areas where the computer vision community has proposed several unsupervised methodologies [62, 63, 64, 65, 61]. We borrow ideas of unsupervised monocular depth estimation, where from two tactile images $I_t$ and $I_{t-n}$, we learn a pose estimator for getting the transform between frames. With the sensor intrinsic $K$, we map image $I_t$ from pixel space to camera plane, translate estimated depth $D_t$, apply transform from $t$ to $t-n$, and transform back to image plane to get $\hat{I}_{t-n}$. We supervised based on the reprojection error, MSE between $I_{t-n}$ and predicted $\hat{I}_{t-n}$. To reconstruct the shear field, we transfer ideas from unsupervised optical flow, where we warp the features of image $I_t$ to $I_{t-n}$ based on the estimated flow and compute a photometric consistency loss that encourages the estimated flow(shear) to align image patches with a similar appearance. This loss is a linear combination of the Charbonnier loss and the structural similarity (SSIM) between $I_{t-n}$ and $\hat{I}_{t-n}$. We also add a smoothness loss that acts as a regularization term, encouraging the shear field to align the boundaries with the visual edges in the tactile image. In Figure 10 we show snapshots of the normal and shear field predictions during sliding trajectories of the DIGIT sensor on YCB and spherical probe objects.

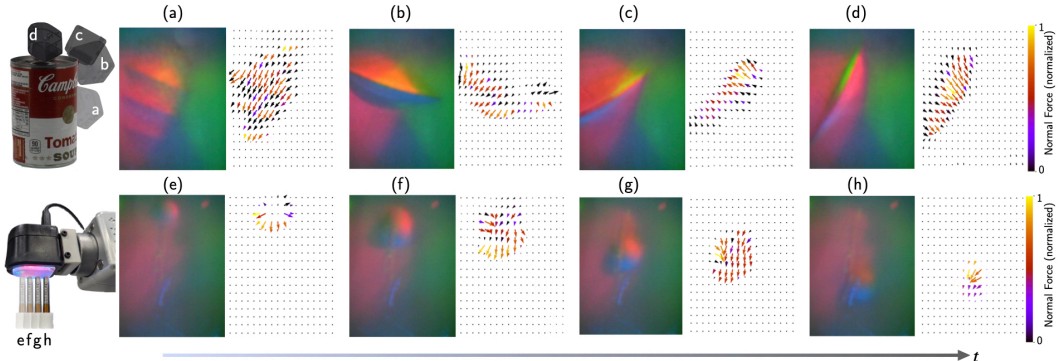

**Figure 10:** Normalized tactile flow (unitless) visualizations using `Sparsh` (`DINO`). Top row shows predicted force field for four key-frames from a representative YCB-Slide trajectory and bottom row shows interaction with the spherical probe. Arrows represent the tangential forces, while the colors depict the normal forces. These visualizations provide directional information about the relative motion of the contact patch. For instance (a) shows torsional motion resulting from rotating along the edge, (b, c, d) show sliding on the edge, (e) shows a diverging field when making contact with a spherical probe, and (f, g, h) show forces produced by sliding the probe top-down.

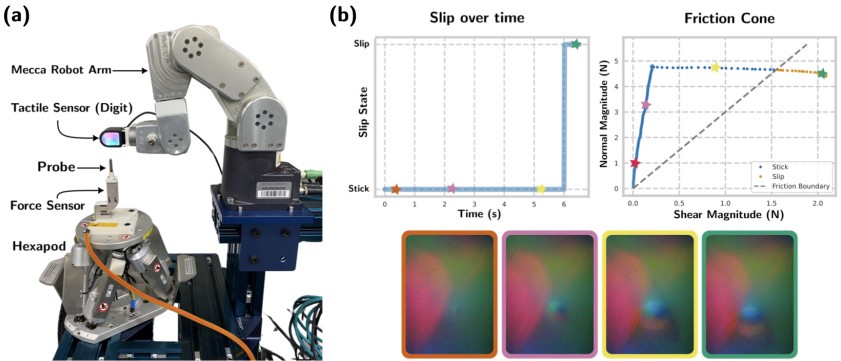

**Figure 11:** (a) Data collection setup for **[T1] Force Estimation** and **[T2] Slip Detection**. The Mecca Robot Arm with DIGIT / Gelsight is pressed against a static probe with random normal force. The arm then slides the sensor over the probe which induces shear forces. (b) Slip states over one representative stroke. When the sensor is pressed against the probe the normal force increases. The gel sensor initially resists sliding due to friction, but gives in, which results in a slight drop in normal force while the magnitude of shear force increases.

### D.5 **[T2] Slip detection**

To collect labeled slip data we perform a normal/shear load test. Using a firmly affixed hemispherical probe on a flat surface, a robot presses the DIGIT sensor toward the probe, applying random normal forces of up to 5N. Upon reaching the target normal force, the robot slides the probe 2mm to a randomly selected position on the sensor surface, allowing us to capture the shear profile with a F/T sensor. To label slip, we rely on the friction cone to identify samples on the sticking and slippage regions. A description of the procedure is illustrated in Figure 11.

As eluded to in Section 3, `Sparsh`'s inference window is approximately 80 milliseconds. This is appropriate since this duration matches the reaction time needed by humans to adjust the grip force when detecting partial slip [53]. We train two heads: one for slip detection and the other for the estimation of normalized force change ($\Delta$). We find empirically that training both heads simultaneously improves slip detection, given their high correlation. The MLP probes are trained with cross-entropy for slip detection and mean absolute error for $\Delta$ force regression as loss functions. Our dataset comprises 125k samples, with only 13% corresponding to slip instances. We reserve 25k samples for evaluating model performance.

Table 7 provides F1-score metrics for all models under different amounts of training data. `Sparsh` (`VJEPA`) outperforms all models, even when trained under low data regimes. In Figure 12 we contrast the predictions over time for a sample trajectory between `Sparsh` (`VJEPA`) and `E2E` models trained

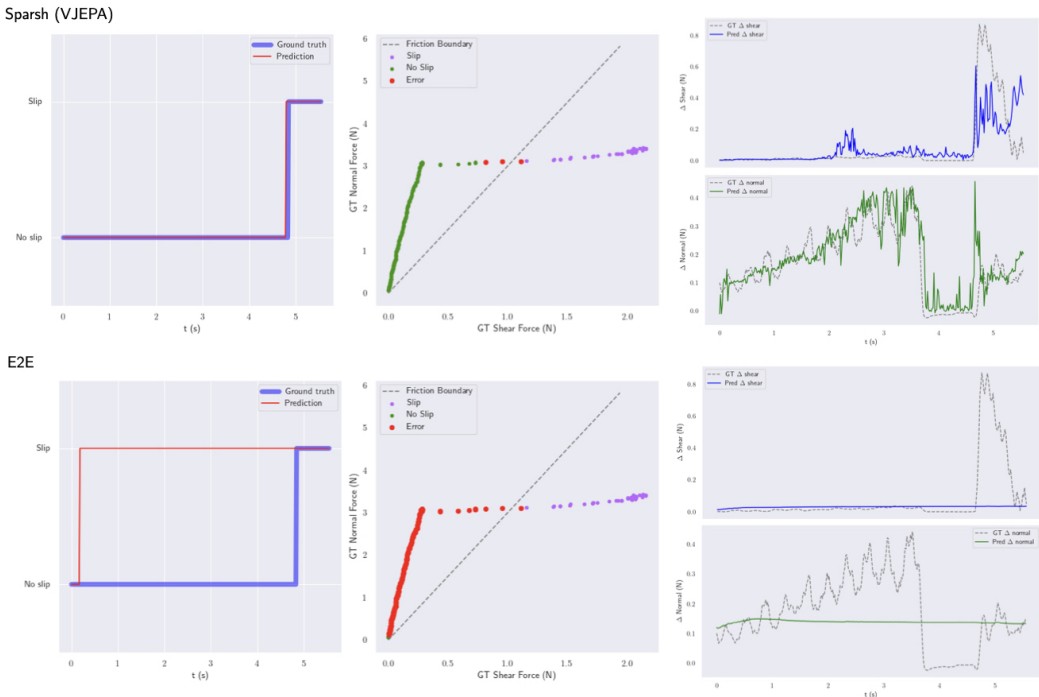

**Figure 12:** Contrast between `Sparsh (VJEPA)` and `E2E` for a test trajectory with a spherical probe sliding on the DIGIT sensor. `Sparsh (VJEPA)`, even though trained only on 33% of the data, can detect slip accurately, which is correlated with its ability to estimate changes in normal and shear forces.

with 33% of the data. Note that for `Sparsh (VJEPA)` the errors are around the friction boundary, where the probe is starting to slide. Also, it is worth noticing that a poor estimation of changes in shear and normal forces is reflected in the accuracy of distinguishing between slip and no-slip. In Figure 13, we illustrate a failure case for `Sparsh (VJEPA)`, as its results do not align with the ground truth. However, it is important to note that slip labeling is prone to errors due to its reliance on an experimental coefficient of friction. Despite the inaccuracies in the friction boundary for this trajectory, `Sparsh (VJEPA)` successfully detects the slip samples.

| Model | Full dataset | 1/3 dataset | 1/10 dataset | 1/100 dataset |
|---|---|---|---|---|
| E2E | 0.767 | 0.238 | 0.299 | 0.214 |
| Sparsh (MAE) | 0.783 | 0.818 | 0.691 | 0.269 |
| Sparsh (DINO) | 0.685 | 0.561 | 0.548 | 0.489 |
| Sparsh (DINOv2) | 0.687 | 0.601 | 0.561 | 0.243 |
| Sparsh (IJEPA) | 0.776 | 0.791 | 0.775 | 0.726 |
| **Sparsh (VJEPA)** | **0.820** | **0.828** | **0.800** | **0.760** |

**Table 7:** Performance of models on slip detection task under different budgets of training data. We use F1 score as metric, given that it ensures the model accurately identifies slip events without favoring the majority class. A high F1 score indicates effective and reliable slip detection.

### D.6  [T3] Pose estimation

We collect a dataset of trajectories with time-synchronized pairs of object pose measurements and sensor observations using an Allegro hand equipped with DIGIT sensors on each finger, mounted on a robot arm. The object was placed on a table and with the palm facing downward, we pressed against it with the fingertips (see Figure 3). We manually perturbed the object's pose by sliding and rotating it under the Allegro fingertips. The pose of the object was tracked using ArUco tags. Given ground truth object pose measurements in the world frame, we preprocess them into relative pose change $(\Delta x, \Delta y, \Delta \theta) \in \mathrm{SE}(2)$ in the sensor frame.

Since we follow a regression-by-classification approach, we discretize the range of motion for each degree of freedom into multiple intervals in Log-uniform space. This allows us to achieve a better

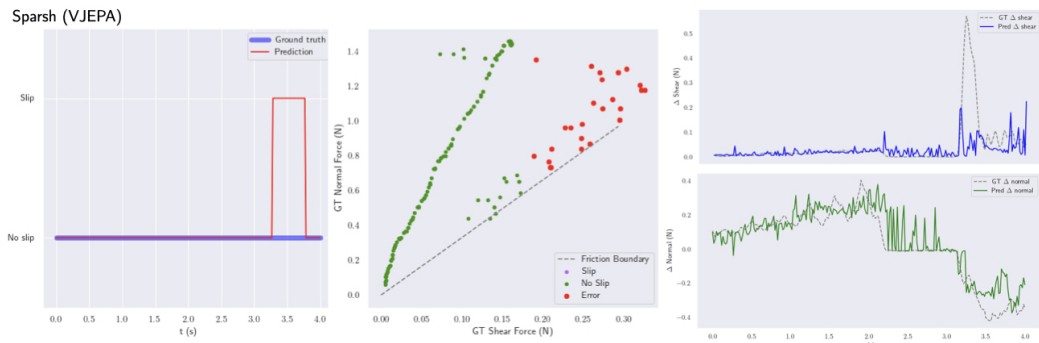

**Figure 13:** Failure case where the ground truth does not reflect slip since it relies on an experimental coefficient of friction. Despite the inaccuracies in the friction boundary for this trajectory, `Sparsh (VJEPA)` successfully detects slip samples.

data distribution across all classes, as most pose changes are concentrated around zero. The strategy of classification-regression is also commonly explored for monocular depth estimation [105].

After attentive pooling, the features are passed to three heads, one for each degree of freedom. Each head is an MLP with two layers, which outputs the probability distribution over 11 classes (pose change bins). In Figure 14 we present the binning as well as the confusion matrices on test data for each degree of freedom, comparing `E2E`, `Sparsh (DINO)` and `Sparsh (IJEPA)` for pose estimation when trained on 33% of the available labeled data. Note that `Sparsh` can accurate distinguish pose changes in a low data regime, while a conventional task-specific approach struggles discerning the differences between adjacent bins, and finally tends to default to zero or maximum relative pose change, losing resolution in estimation.

Figure 15 shows a test trajectory over time with its ground truth labels. The colors on the plot represent the class agreement between the pose decoders trained with `Sparsh (DINO)` (using 33% of the data) and the ground truth. Darker colors indicate no error, while brighter colors indicate greater misclassification. In Table 8 we report for each model accuracy in pose estimation over 630 test samples and 95% confidence interval.

| Model | Full dataset | 1/3 dataset | 1/10 dataset | 1/100 dataset |
|---|---|---|---|---|
| E2E | 0.812 [0.811, 0.813] | 0.245 [0.244, 0.247] | 0.162 [0.160, 0.164] | 0.162 [0.160, 0.164] |
| Sparsh (MAE) | 0.896 [0.896, 0.897] | 0.719 [0.718, 0.721] | 0.417 [0.414, 0.420] | 0.223 [0.221, 0.225] |
| **Sparsh (DINO)** | **0.913 [0.912, 0.914]** | **0.834 [0.832, 0.836]** | **0.460 [0.457, 0.461]** | **0.242 [0.240, 0.245]** |
| Sparsh (DINOv2) | 0.665 [0.658, 0.673] | 0.565 [0.559, 0.570] | 0.411 [0.408, 0.415] | 0.210 [0.209, 0.211] |
| Sparsh (IJEPA) | 0.851 [0.850, 0.852] | 0.601 [0.599, 0.603] | 0.323 [0.321, 0.325] | 0.212 [0.210, 0.215] |
| Sparsh (VJEPA) | 0.856 [0.854, 0.857] | 0.648 [0.646, 0.651] | 0.368 [0.367, 0.370] | 0.228 [0.225, 0.231] |

**Table 8:** Accuracy and 95% confidence interval for pose estimation task following the regression-by-classification paradigm. Relative pose between object and ring finger. Metrics computed over 630 test samples.

### D.7 [T4] Grasp stability

We use the Feeling of Success dataset [8], which contains data from a pair of GelSight sensors (with markers) attached to a jaw gripper (left and right fingers). The goal is to determine the success or the failure of the grasp attempt.

We pass to the SSL model the 'before' and 'during' as tactile history. We create our randomized split with all objects, using approximately 8k grasps for training and the remaining 1.3k grasps for

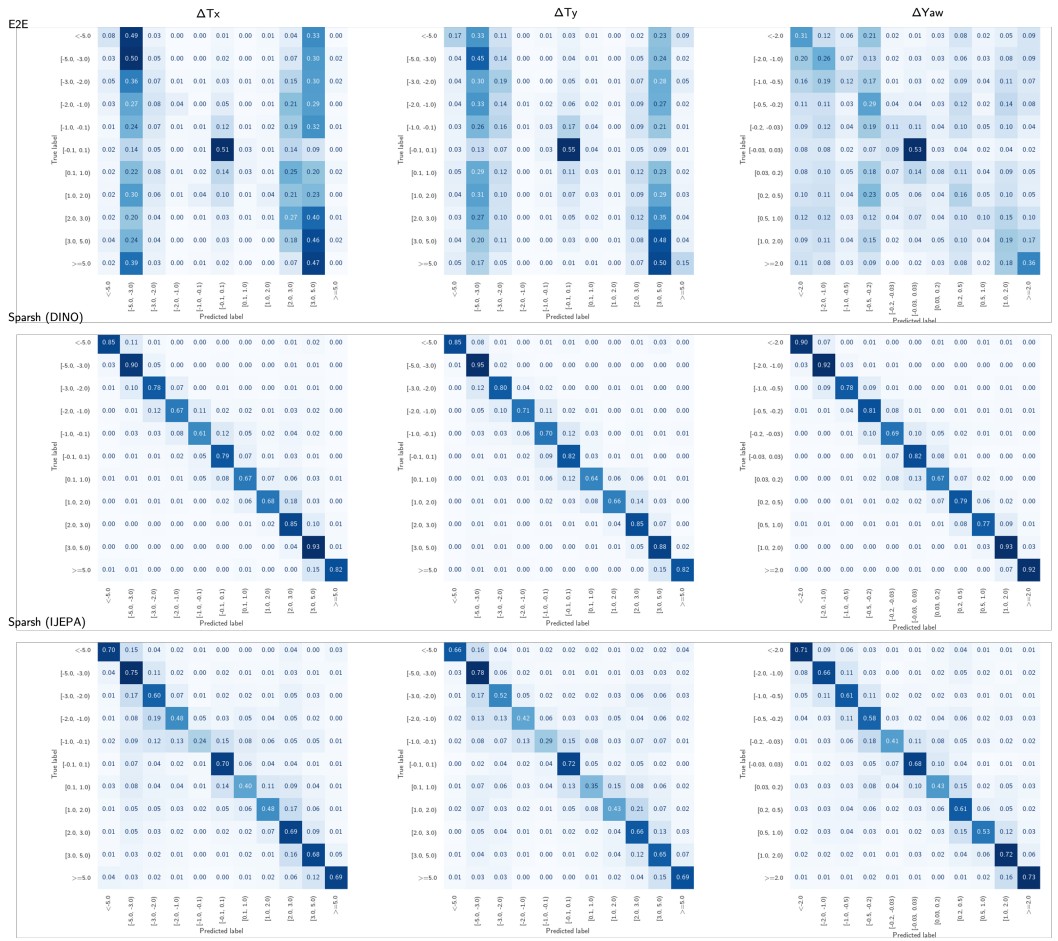

**Figure 14:** Confusion matrix on test data for $\Delta T_x$, $\Delta T_y$, $\Delta$Yaw for E2E, Sparsh (DINO) and Sparsh (IJEPA) trained on 33% of the available labeled data. The test dataset consist of 630 samples.

evaluation. Using attentive probing, we freeze Sparsh and train a 2-layer MLP with two output units for grasp success classification.

In Table 9 report the accuracy for binary classification to compare the performance of the models across different training budgets, including a 95% confidence interval. Figure 16 shows the confusion matrices on test samples for E2E, Sparsh (DINO) and Sparsh (IJEPA) trained on a 33% of labeled data.

### D.8 [T5] Textile recognition

This tasks allows to study the capabilities of the representations for semantic understanding of the contact, as in recognizing the type of textile that is being touched by the sensors. We use the task definition and the data set introduced in [58]. This data set contains 4467 short video clips (10-25 frames), of a robot with a GelSight (markers) mounted parallel gripper grasping several types of clothing, across 20 textile classes, such as leather, cotton, polyester, etc.

We follow the train-test split provided in the metadata of the dataset. Using attentive probing, we freeze Sparsh and train a 2-layer MLP with 20 output units for textile classification. In Table 10 and Figure 17(c) we report the accuracy for multiclass classification, comparing the performance of the models in different training budgets.

### D.9 [T6] Bead maze

The goal in bead maze is to guide the bead along the wire, as shown in Figure 3. We don't rely on vision for hand-eye coordination, making the task fundamentally tactile since forces in the fingers indicate whether the bead is moving smoothly or encountering resistance. In our setup, we use

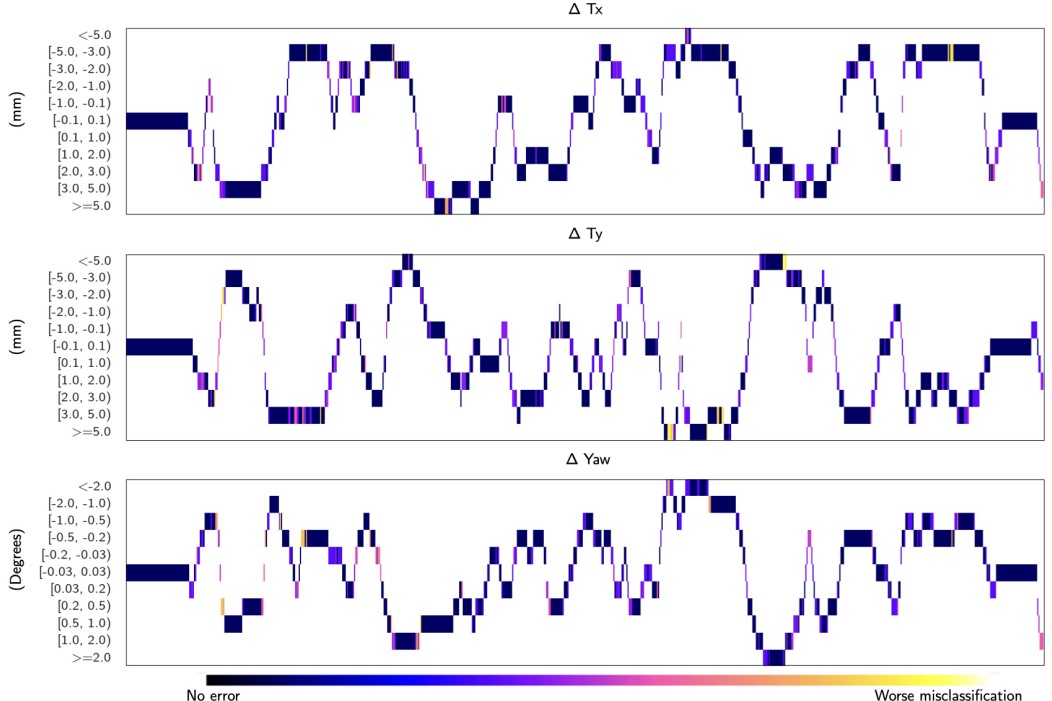

**Figure 15:** Ground truth relative pose classes for $T_x$, $T_y$, and Yaw for a test trajectory. The colormap represents the class agreements between the ground truth and the pose decoder, with darker colors indicating no error and brighter colors indicating greater misclassification.

| Model | Full dataset | 1/3 dataset | 1/10 dataset | 1/100 dataset |
|---|---|---|---|---|
| E2E | 0.784 [0.783, 0.785] | 0.725 [0.722, 0.728] | 0.682 [0.680, 0.684] | 0.478 [0.472, 0.482] |
| Sparsh (MAE) | 0.815 [0.813, 0.817] | 0.696 [0.691, 0.702] | 0.764 [0.761, 0.768] | 0.466 [0.461, 0.471] |
| Sparsh (DINO) | 0.780 [0.777, 0.782] | 0.706 [0.702, 0.710] | 0.773 [0.772, 0.775] | 0.473 [0.467, 0.479] |
| Sparsh (DINOv2) | 0.770 [0.767, 0.771] | 0.770 [0.768, 0.772] | 0.699 [0.697, 0.701] | 0.543 [0.539, 0.546] |
| **Sparsh (IJEPA)** | 0.802 [0.800, 0.804] | **0.782** **[0.779, 0.784]** | **0.768** **[0.766, 0.770]** | **0.598** **[0.597, 0.601]** |
| Sparsh (VJEPA) | **0.809** **[0.805, 0.813]** | 0.702 [0.700, 0.704] | 0.743 [0.740, 0.746] | 0.523 [0.519, 0.527] |

**Table 9:** Accuracy and 95% confidence interval for grasp stability classification over different budget sizes of training data, using Feeling of Success dataset. Results over 1.3k grasps.

a Franka arm with a robotic hand mounted on the wrist and DIGIT sensors on the fingers. To collect demonstrations for training the policy, we start the task with the bead grasped between the thumb and index fingers and move the arm to guide the bead along the wire. We collect 30 demonstrations on different maze patterns with mix of VR-based and manual kinesthetic-based teleoperation, corresponding to a total of ∼34k training pairs of tactile images and robot joint angles.

For training the policy, we adapt Diffusion Policy [71] to our problem setting. Given a small history of tactile images $(\ldots, \mathbf{z}_{t-1}, \mathbf{z}_t)$, and robot proprioception $(\ldots, q_{t-1}, q_t)$, we train the policy to predict changes in joint angles as actions $\mathbf{a} \triangleq (\Delta q_t, \Delta q_{t+1}, \ldots)$; $\Delta q \in \mathbb{R}^7$, instead of position control. Following the guidelines in Diffusion Policy, we use an observation horizon of 2 and an action prediction horizon of 8. We adhere to the official implementation for policy architecture and training hyper-parameters. For conditioning on tactile input, we modify the CNN encoder from Diffusion

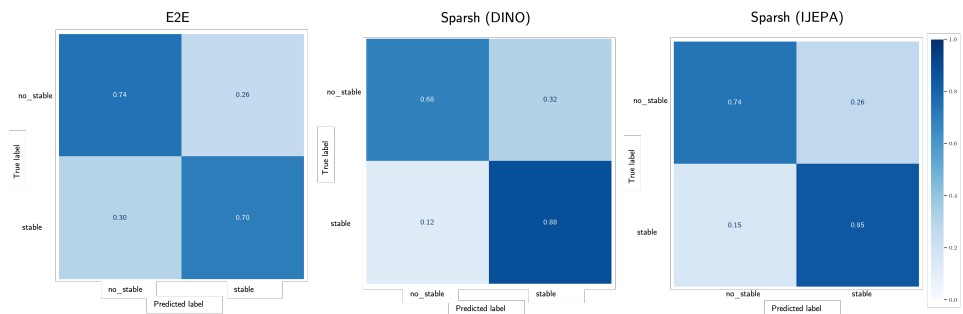

**Figure 16:** Confusion matrix on test data for grasp stability, comparing `E2E`, `Sparsh (DINO)` and `Sparsh (IJEPA)` trained on 33% of the available labeled data. The test dataset consist of 1.3k grasps.

| Model | Full dataset | 1/3 dataset | 1/10 dataset | 1/100 dataset |
|---|---|---|---|---|
| E2E | 0.437 | 0.365 | 0.373 | 0.171 |
| **Sparsh (MAE)** | **0.599** | **0.588** | **0.527** | **0.330** |
| Sparsh (DINO) | 0.527 | 0.520 | 0.463 | 0.264 |
| Sparsh (DINOv2) | 0.544 | 0.536 | 0.469 | 0.288 |
| Sparsh (IJEPA) | 0.506 | 0.478 | 0.399 | 0.217 |
| Sparsh (VJEPA) | 0.580 | 0.545 | 0.507 | 0.285 |

**Table 10:** Accuracy for textile classification over 20 classes using GelSight with markers dataset under different budget of labeled data. Results over 26k tactile images, where accuracy of chance is 0.05.

Policy and replace it with `Sparsh` backbones with fixed parameters. For training an end-to-end policy, the encoder corresponds to a ViT-Base encoder with randomly initialized weights.

For each method, we evaluate the learned policies over a set of 10 randomized novel starting locations on the maze and we measure distance traversed (in cm) before failure. In Table 11, we report mean and variance of distance traversed comparing `Sparsh` (pre-trained only and pre-trained then fully fine-tuned) against `E2E`. All models use 50 demonstrations for training the policy via imitation learning. We find that policies using `Sparsh` representations outperform `E2E` by $\sim 20 - 53\%$. Most failure cases across methods are due to the bead getting stuck on the maze or the bead falling out of the robot hand. While prior work such as diffusion policy suggest that frozen pre-trained models may hurt imitation learning due to domain mismatch, we do not observe significant gains from fine-tuning in this application. Leveraging pre-trained models in imitation learning is an active area of research, however these results demonstrate the impact of `Sparsh` touch representations for robot applications.

| (cm) | Sparsh (DINO) | Sparsh (IJEPA) | Sparsh (MAE) | E2E |
|---|---|---|---|---|
| Pre-trained | $10.80 \pm 3.68$ | $9.4 \pm 3.1$ | $10.2 \pm 4.9$ | $6.70 \pm 1.67$ |
| Fine-tuned | $8.45 \pm 3.21$ | $10.02 \pm 5.37$ | $11.25 \pm 3.85$ | $6.70 \pm 1.67$ |

**Table 11:** Mean and variance of distance traversed (in cm) before failure for policies based on `Sparsh` and `E2E`. Results over 10 randomized novel starting locations on the bead maze.

In Table 12 we report to position error of `E2E`, `Sparsh (DINO)` and `Sparsh (IJEPA)` with respect to test demonstrations on an unseen maze, highlighting the fidelity of `Sparsh (DINO)` and `Sparsh (IJEPA)` to follow a similar trajectory. Nevertheless, this doesn't necessarily transfer to real-world performance, since the locality of the observations and predictions make the errors in the adjusted joint angles to compound fast, which results in unforeseen collisions and the subsequent lose of the grasp. In an overfitting setting, training a policy for a single maze, policies using `Sparsh (DINO)` and `Sparsh (IJEPA)` are able to complete almost 30% of the maze on the real robot. However, it is expected an specialist policy trained end-to-end to perform better in the overfitting setting. Experimentally, we found than an `E2E` policy trained for a single maze is able to complete almost 80% of the maze running on the real robot.

In Table 13 we summarize the performance of `Sparsh` across the benchmark. We find that with respect to an `E2E` approach, with `Sparsh` we can achieve an improvement of 98.75% on average. `Sparsh (DINO)` and `Sparsh (IJEPA)` are in general the best models across the board, showing the

benefits of learning touch representations in latent space. Sparsh (MAE), which relies on pixel space supervision, is still competitive, although it was not evaluated on the policy task.

| Model | Full dataset | 1/2 dataset | 1/10 dataset |
|---|---|---|---|
| Sparsh-(E2E) | 8.46 [7.61, 9.32] | 7.14 [6.26, 8.05] | 9.80 [8.78, 10.82] |
| Sparsh-(DINO) | 5.54 [4.90, 6.17] | 5.98 [5.29, 6.67] | 5.71 [5.13, 6.29] |
| Sparsh-(IJEPA) | 5.47 [4.82, 6.13] | 5.72 [5.05, 6.40] | 5.46 [4.82, 6.10] |

**Table 12:** Position error (mm) and 95% confidence interval for the Bead Maze task. We compare the ground truth trajectory from a test demonstration in an unseen maze against the compounded trajectory from the predicted delta joint angles from each policy.

| Task | Best SSL vs E2E | DINO vs IJEPA | MAE vs Best | VJEPA vs Best |
|---|---|---|---|---|
| Force estimation (DIGIT) | 28.31% | 26.67% | −4.38% | −27.96% |
| Force estimation (GelSight) | 59.74% | 32.41% | 1.72% | −64.23% |
| Slip detection | 242.70% | 29.08% | −1.21% | 0.00% |
| Pose estimation | 235.89% | −37.91% | −13.81% | −22.33% |
| Grasp stability | 5.14% | 8.45% | −10/17% | −7.83% |
| Bead maze | 19.72% | −5.26% | - | - |
| *Average* | **98.75**% | 8.91% | −5.57% | −24.47% |

**Table 13:** Performance of Sparsh across TacBench and comparison between SSL approaches.

# E  Sparsh ablations

## E.1  TacBench evaluations via fine-tuning

Fine-tuning the Sparsh encoders is another method of assessing the quality of pre-trained representations. Fine-tuning can potentially enhance performance in downstream tasks when the pre-trained model lacks task-relevant information.

We evaluated both the full and partial fine-tuning of Sparsh on TacBench. In full fine-tuning, all encoder parameters are updated through task supervision. In partial fine-tuning, we update only the last transformer block of the encoder. Figure 17 shows the fine-tuning results in the benchmark with varying amounts of labeled data. Notably, models pre-trained in latent space (DINO, I-JEPA, V-JEPA) perform better in downstream tasks when fully fine-tuned, especially in regression tasks like force and pose estimation. For example, Figure 17(a) illustrates that errors in force estimation are significantly lower with full fine-tuning, even with only 33% and 10% labeled data. Full fine-tuning also enhances performance in classification tasks such as slip detection, grasp stability, and textile classification, as shown in Figures 17(b,c). Adding in-domain data to the encoder reduces performance gaps in the benchmark between Sparsh (DINO), Sparsh (IJEPA), and Sparsh (VJEPA). However, this method is less effective for the Sparsh (MAE) model, which is trained in pixel space. We hypothesize that MAE weights are potentially more brittle when compared to other SSL models which enjoy a wider basin of minima due to weight updates via exponential moving average. In contrast, partial fine-tuning offers minor improvements, aligning closely with the performance of frozen models.

## E.2  Sparsh ViT-small and performance

We train Sparsh for all SSL approaches decreasing the model capacity by using a transformer ViT-small. This let us study the effect of the dimensionality of the touch representations on downstream tasks, from 768 with ViTbase to 384 with ViTsmall. We follow the same training procedure explained in Appendix C.

We evaluate Sparsh-vitsmall across TacBench. In Figure 17 we report the performance of each task for different budgets of labeled data following the attentive probing protocol. Reducing the dimensionality of the representations do plays an important role for some tasks. Regression-like tasks such as [T1] Force estimation (see Figure 17a) exhibit a decrease in performance when reducing the

capacity of the encoders, specially when the downstream tasks needs to be trained under a limited number of labeled data. For instance, Sparsh (DINO) increases the force estimation error by $74\%$ for DIGIT and $50.3\%$ for GelSight Mini when using representations from Sparsh-vitsmall and training the downstream tasks with $33\%$ of labeled data. The decrease in performance is also observe in regression-by-classification tasks, as in [T3] Pose estimation. With Sparsh-vitsmall all models perform very similar but losing $20\%$ accuracy even when training the downstream tasks with the full labeled dataset. Nevertheless the performance is still better than an E2E model with a vitbase encoder.

In general for classification tasks in the benchmark like [T2] Slip detection, [T4] Grasp stability and [T5] Textile recognition, there is no major effect of reducing the capacity of the encoder. The drop in performance is only significant when training the downstream task with the lowest amount of training data, $1\%$ in our experiments.

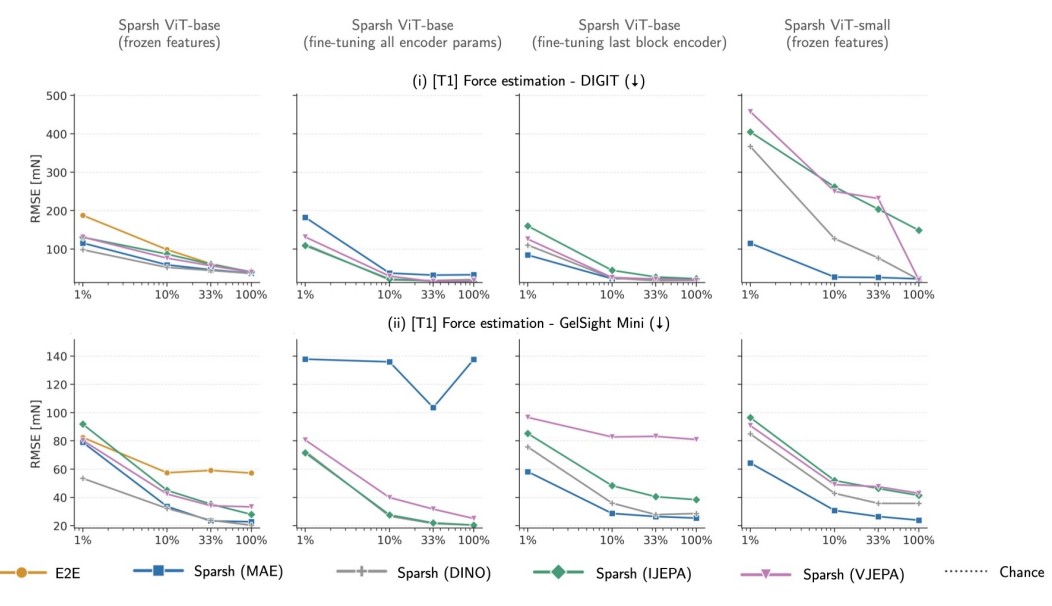

(a) Additional evaluations for T1 force estimation.

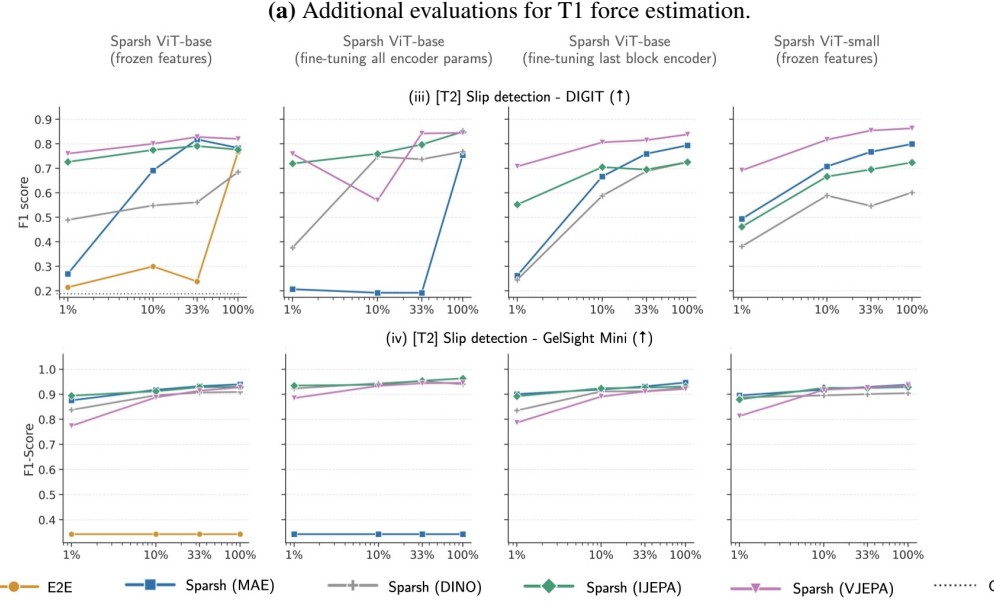

(b) Additional evaluations for T2 slip detection.

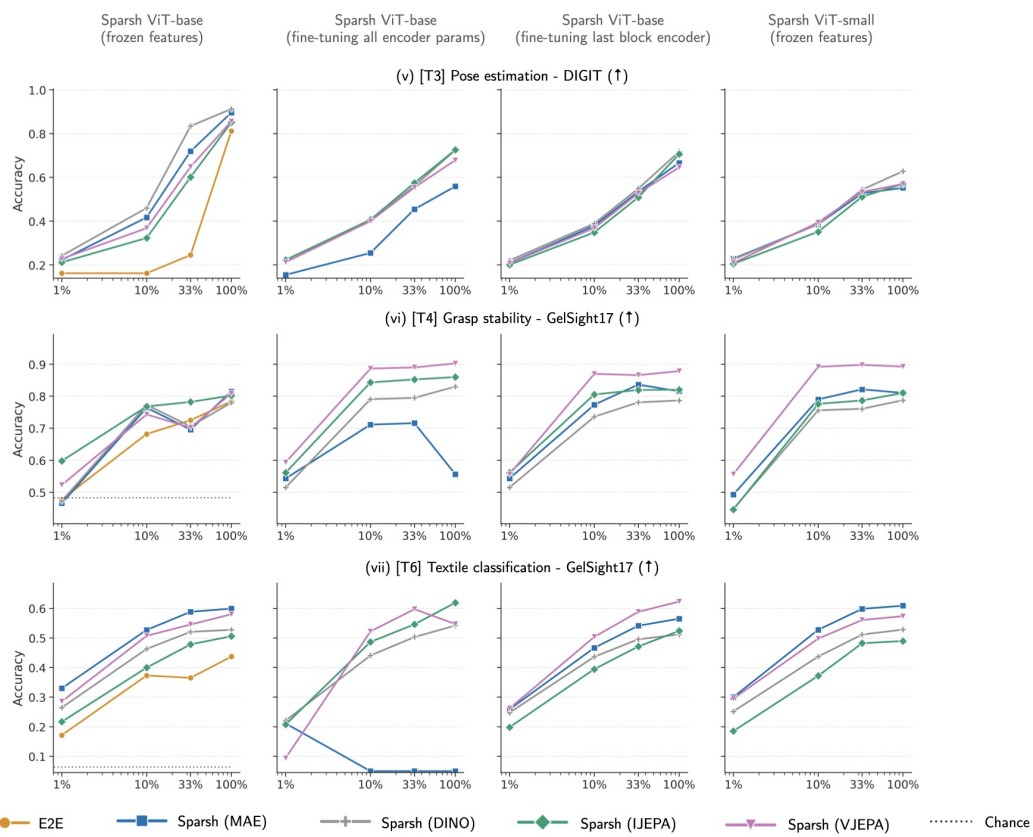

**(c)** Additional evaluations for the perception tasks, T3 pose estimation (top), T4 grasp stability (middle) and T6 textile classification (bottom).

**Figure 17:** Additional evaluations of `Sparsh` representations on `TacBench`. We compare frozen `Sparsh` ViT-base (most left), `Sparsh` fully and partially fine-tuned (middle) and finally (most right) `Sparsh` ViT-small to gauge the effect of reducing the dimensionality of the representations.

### E.3 Sparsh and cross-sensory representation

Since `Sparsh` is trained on multiple GelSight-like data, we investigate whether SSL training enables cross-sensory representations or if it helps downstream tasks trained for one sensor quickly adapt to another. To study this, we use as a baseline the decoder trained for [T5] Textile recognition, which was supervised with labeled data from GelSight with markers.

We collect new data using a DIGIT sensor for 10 out of the 20 textiles. Our dataset contains 11 samples for both training and testing. We load the trained decoders for [T5] using `Sparsh` (DINO) and E2E and perform zero-shot evaluation as well as 1-shot, 5-shot, and 10-shot training and subsequent evaluation using the DIGIT data. Table 14 reports accuracy on 110 samples of test data. Zero-shot evaluation with DIGIT performs close to chance, while with very few samples (10-shot) `Sparsh` (DINO) classifier quickly adapts and significantly outperforms E2E. This experiment empirically demonstrates the value of cross-sensor representations.

|  | zero-shot | 1-shot | 5-shot | 10-shot |
|---|---|---|---|---|
| Sparsh (DINO) | 9.1 | 19.1 | 28.2 | 61.8 |
| E2E | 3.6 | 0.0 | 15.5 | 10.9 |

**Table 14:** Accuracy of $n$-shot evaluation of [T5] Textile recognition on DIGIT data to study how `Sparsh` facilitates cross-sensory adaptation to the downsntream task.

