# OpenReview forum: "Sparsh: Self-supervised touch representations for vision-based tactile sensing"
_robot-learning.org/CoRL/2024/Conference — CoRL 2024_

### Official Review · Reviewer_B6vw · 2024-07-20

**Originality:** 4
**Technical Quality:** 3
**Clarity Of Presentation:** 2
**Potential Impact:** 3
**Recommendation:** 3
**Confidence:** 5

**Review:**

**Strengths:**
1. The paper addresses an important problem and its initial parts are very well written which introduces the significance of the problem.
2. The authors makes their contribution clear.

**Weaknesses:**
1. **Method Section:** The method section is not adequately written. It is understandable that the approach is utilizing the existing method, but still the authors must mention some details of I/V-JEPA (along with loss function) in the main paper. I would suggest the authors to refer to one of their mentioned paper [1], which could serve as a useful template for enhancing the clarity and depth of the method section (as it also leverages on existing method but a detailed description of the method is provided).

2. **Data Bias:** Given that the datasets are derived from multiple sensors and vary significantly in the number of images, there can be some potential bias. It is important to investigate whether the model shows a preference for certain sensors, thereby learning better from some sensors than others.

3. **Cross-Sensory representation:** The concept of learning "cross-sensory representations" requires further clarification. Specifically, it is unclear whether there is an actual transfer of representations between sensors. For instance, considering the Touch-and-Go [2] dataset, which primarily contains data collected in the wild using the Gelsight sensor. Can the method recognise in a one-shot scenario with a different type of tactile sensor, such as Digit, when presented with a similar object like Grass? This analysis is essential for understanding the generalizability of the proposed method across different tactile sensors.

**Minor Details:**
1. The image reference in line 717 is missing.

**Suggestions:**
1. I would suggest to include these following recent papers in SSL domain [3,4] and this very recent work which is very similar to yours [5,6].
2. It is hard to see the accuracy and F1 Score from the graphs in the Figure 4. I would highly suggest to include the table with bold text as the best result (as usually done).

**References:**

[1] Yang, Fengyu, et al. "Binding touch to everything: Learning unified multimodal tactile representations." Proceedings of the IEEE/CVF Conference on Computer Vision and Pattern Recognition. 2024.

[2] Yang, Fengyu, et al. "Touch and go: Learning from human-collected vision and touch." arXiv preprint arXiv:2211.12498 (2022).

[3] Kerr, Justin, et al. "Self-supervised visuo-tactile pretraining to locate and follow garment features." arXiv preprint arXiv:2209.13042 (2022).

[4] Dave, Vedant, Fotios Lygerakis, and Elmar Rueckert. "Multimodal visual-tactile representation learning through self-supervised contrastive pre-training." arXiv preprint arXiv:2401.12024 (2024).

[5] Fu, L., Datta, G., Huang, H., Panitch, W. C. H., Drake, J., Ortiz, J., ... & Goldberg, K. (2024). A touch, vision, and language dataset for multimodal alignment. arXiv preprint arXiv:2402.13232.

[6] Zhao, Jialiang, et al. "Transferable Tactile Transformers for Representation Learning Across Diverse Sensors and Tasks." arXiv preprint arXiv:2406.13640 (2024).

**Quality Of The Limitations Section:**

2

**Questions For Rebuttal:**

**Questions:**
1. Is any sensor-specific detail provided to your encoder during training, similar to the approach in [1]? If not, why not? Do you think it can improve the training and generate sensor dependent representations?
2. In the results section (Section 6.2), it is mentioned that classification relies solely on single finger data. How is it possible to predict grasping success or failure from just this data? Does this imply that the dataset has some inherent linear relationship with grasping success, such as greater deformation correlating with higher success rates?

**Robotics Focus:**

4

**Summary Of Paper:**

The paper presents Sparsh, a self-supervised learning model family for vision-based tactile sensors, addressing the limitations of task-specific handcrafted models. By pre-training on over 460k tactile images with masking and self-distillation in pixel and latent spaces, Sparsh eliminates the need for custom labels. The authors introduce TacBench, a benchmarking suite for evaluating tactile properties and manipulation across different sensors and models. They found out that DINO and I-/V-JEPA are the most dominant approaches.

**Summary Of Recommendation:**

The paper tries to solve a completely reasonable problem and it can also prove to be very effective in this domain. However, certain parts of the paper requires revision as described in the Weaknesses. As of now, I am inclined towards Weak Reject but I  am open to reconsider my score if the authors provide a satisfactory rebuttal addressing these concerns.

---

### Official Review · Reviewer_EZmQ · 2024-07-21
**Self-supervised touch representation**

**Originality:** 3
**Technical Quality:** 4
**Clarity Of Presentation:** 4
**Potential Impact:** 3
**Recommendation:** 3
**Confidence:** 4

**Review:**

The paper's main strengths are: 1. The proposed framework is well described and explained with words and figures. The paper employs a comprehensive self-supervised learning (SSL) approach, which is rigorously tested and validated across multiple tasks and sensors. 2. Extensive Evaluation: The use of TacBench, a standardized benchmark, allows for a thorough evaluation of the models across five distinct touch-centric tasks. This ensures the results are robust and generalizable.

The main weaknesses of the paper are:
1. Some parts of the paper are dense with technical terms, which could be challenging for readers who are not specialists in the field. Refer to the list of issues for more detailed questions.
2. The reliance on large datasets and sophisticated models suggests high computational requirements. This paper uses 8 Nvidia A100 GPUs, which might limit accessibility for some researchers.
3. There is a lack of a more intuitive illustration of the learned tactile representation. Besides the resultant higher performance thanks to the proposed models, further discussions on the learned representation would greatly help contribute to the current understanding of vision-based tactile sensing.

**Quality Of The Limitations Section:**

3

**Questions For Rebuttal:**

Issues:
1.	Are the results presented in Figure 1 (middle) the same as those in Figure 4? What is the amount of labeled data in Figure 1? This information should be included in the caption as the outperforming only exists under a limited labeled data budget.
2.	As a benchmark, the end-to-end (E2E) approach is not well explained, and the reviewer failed to find the differences between E2E and the proposed Sparsh models in the paper. In lines 160-162, it seems E2E has identical model capacity. What are the differences?
3.	In line 148, predicting the stability of a grasp should be T4 instead of T2.
4.	In line 277, what is behavior closing?

**Robotics Focus:**

4

**Summary Of Paper:**

The paper introduces Sparsh, a family of self-supervised learning (SSL) models that support various vision-based tactile sensors. It addresses the challenge of task and sensor-specific models by pre-training on a large dataset of over 460,000 tactile images, enabling general-purpose touch representations without custom labels. The paper also presents TacBench, a standardized benchmark with five touch-centric tasks to evaluate these models. Evaluations show that Sparsh models significantly outperform task-specific end-to-end training, with Sparsh (DINO) and Sparsh (IJEPA) demonstrating the most competitive performance. Additionally, the paper curates a unified dataset from new and existing sources, enhancing the scope and applicability of tactile sensing research.

**Summary Of Recommendation:**

The paper presents a practical approach for touch representations. The idea is interesting to the community and the presentation is well organized. Hence, I recommend accepting the paper.

---

### Official Review · Reviewer_brZ3 · 2024-07-21
**Self-supervised representation learning for visuo-tactile sensors**

**Originality:** 3
**Technical Quality:** 3
**Clarity Of Presentation:** 3
**Potential Impact:** 3
**Recommendation:** 3
**Confidence:** 5

**Review:**

**Strengths**

- The paper is well-written and easy to follow.
- It addresses the challenge of learning effective tactile representations for visuo-tactile sensors that are useful for various downstream tasks, including force estimation, slip detection, object pose-change estimation, and grasp stability.
- The results suggest that pretraining with self-supervised objectives helps derive latent features that make downstream task learning more sample efficient.

**Weaknesses**

- It is unclear how the learned features are used for downstream task learning. Details about the decoder architectures for each task are missing. What is the dimensionality of the learned feature representation? Is this dimensionality an important factor?
- It appears that the learned representation is frozen while a decoder head is learned for each task. Is this understanding correct? What happens if the full model is unfrozen and fine-tuned at the end of training? Does task performance improve with additional fine-tuning of the full model?
- The work does not demonstrate that the representation is useful for robotic applications beyond reporting evaluation metrics on train/test splits. It does not address how planners or policies may perform on the tasks. For example, Section 7 reports results for the bead maze task as trajectory errors rather than a measure of task success, such as distance to goal or fraction of path before failure. Using metrics like trajectory errors assumes that the reference trajectory is an optimal solution, but the demonstration collection medium is likely to be suboptimal.
- It is unclear how well the representation generalizes to new objects or new sensors, as evaluation is done by splitting the dataset into training/test sets. Additional generalization analysis would be valuable.

**Quality Of The Limitations Section:**

2

**Questions For Rebuttal:**

See Review above.

**Robotics Focus:**

3

**Summary Of Paper:**

This work presents a self-supervised learning (SSL) approach to touch representation learning for vision-based tactile sensors, addressing challenges in task-specific models and data collection. The authors introduce Sparsh, a suite of three SSL models (MAE, DINO, and IJEPA) pre-trained on over 460k tactile images from three types of visuo-tactile sensors using masking and self-distillation, eliminating the need for custom labels. They also develop TacBench, a standardized benchmarking platform with five tasks. Results evaluated by splitting datasets into training/test sets show that SSL pre-training outperforms end-to-end training by an average of 98.7% on TacBench task datasets, with Sparsh (DINO) and Sparsh (IJEPA) models demonstrating superior performance.

**Summary Of Recommendation:**

The work addresses the interesting problem of visuo-tactile feature learning for sample-efficient task learning. However, some critical details about how to use the learned representation for task learning are missing. Additional experiments demonstrating how the proposed method performs in manipulation planning would highlight its potential impact.

---

### Author Rebuttal · Authors · 2024-08-12

Based on reviewer feedback, we attach our revised manuscript + appendix. For clarity these updates in the paper are highlighted in **blue** color. Major updates include:

* [B6vw] details of SSL methods, [brZ3] details on using decoders and decoder architectures, and new results with ViT-small architecture comparing impact of dimensionality of the representation
* [brZ3] new experiments evaluating complete and partial fine-tuning of pre-trained representations
* [brZ3] new comprehensive real world policy evaluations on [T5] Bead maze task
* [brZ3] clarification on generalization to novel sensors and objects, and [B6vw] new experiments with GelSight mini on [T2] Slip detection, and new task [T6] Textile recognition with zero to few shot cross-sensor transfer results clarifying data bias and generalization
* [EZmQ] details on interpreting learned representations

We thank all the reviewers for their valuable inputs!

---

### Decision · Program_Chairs · 2024-09-04

**Decision:**

Accept

**Comment:**

**Before Rebuttal**

Strengths. The paper is well motivated and easy to follow (albeit with some issues in the method section), with a clear technical contribution. There is strong evidence that the proposed method for self-supervised learning of tactile representations is superior to end-to-end learning. There are extensive experiments across multiple sensors and multiple challenges in tactile sensing. The paper also introduces a new benchmark, TacBench.

Weaknesses. Some aspects of the method are difficult to follow. Experiments are limited to standard machine learning evaluations on test/train datasets, with a lack of experiments studying downstream policy learning on realistic tasks. There is insufficient information on the learned features and how they are used for task learning. It is unclear if the learned representations would transfer well to novel objects. There is a lack of depth in experiments regarding the different sensor types, such as generalisation to novel sensors or sensor bias in the dataset.

---

**After Rebuttal**

Following the reviews, authors replied to the reviewers and addressed some of the concerns raised. Authors also provided a revised paper, including some new experiments. These responses and new material were received well by the reviewers, with two of the reviewers then upgrading their recommendation. Following the rebuttal, the AC and reviewers had a discussion, where all three reviewers now recommend acceptance of the paper. There is agreement that, whilst the paper lacks in distinct algorithmic novelty, the experiments, results, and new benchmark, are sufficiently strong that this paper would be interesting and useful to those in the CoRL community working on tactile sensing.